# Sparse Learning with CART

**Jason M. Klusowski**
Department of Operations Research &
Financial Engineering
Princeton University
Princeton, New Jersey 08544
`jason.klusowski@princeton.edu`

## Abstract

Decision trees with binary splits are popularly constructed using Classification and Regression Trees (CART) methodology. For regression models, this approach recursively divides the data into two near-homogenous daughter nodes according to a split point that maximizes the reduction in sum of squares error (the impurity) along a particular variable. This paper aims to study the statistical properties of regression trees constructed with CART methodology. In doing so, we find that the training error is governed by the Pearson correlation between the optimal decision stump and response data in each node, which we bound by constructing a prior distribution on the split points and solving a nonlinear optimization problem. We leverage this connection between the training error and Pearson correlation to show that CART with cost-complexity pruning achieves an optimal complexity/goodness-of-fit tradeoff when the depth scales with the logarithm of the sample size. Data dependent quantities, which adapt to the dimensionality and latent structure of the regression model, are seen to govern the convergence rates of the prediction error.

## 1 Introduction

Decision trees are the building blocks of some of the most important and powerful algorithms in statistical learning. For example, ensembles of decision trees are used for some bootstrap aggregated prediction rules (e.g., bagging [2] and random forests [3]). In addition, each iteration of gradient tree boosting (e.g., TreeBoost [7]) fits the pseudo-residuals with decision trees as base learners. From an applied perspective, decision trees have an appealing interpretability and are accompanied by a rich set of analytic and visual diagnostic tools. These attributes make tree-based learning particularly well-suited for applied sciences and related disciplines—which may rely heavily on understanding and interpreting the output from the training algorithm. Although, as with many aspects of statistical learning, good empirical performance often comes at the expense of rigor. Tree-structured learning with decision trees is no exception—statistical guarantees for popular variants, i.e., those that are actually used in practice, are hard to find. Indeed, the recursive manner in which decision trees are constructed makes them unamenable to analysis, especially when the split protocol involves *both* the input and output data. Despite these challenges, we take a step forward in advancing the theory of decision trees and aim to tackle the following fundamental question:

*When do decision trees adapt to the sparsity of a predictive model?*

To make our work informative to the applied user of decision trees, we strive to make the least departure from practice and therefore focus specifically on Classification and Regression Tree (CART) [4] methodology—by far the most popular for regression and classification problems. With this methodology, the tree construction importantly depends on both the input and output data and is therefore *data dependent*. This aspect lends itself favorably to the empirical performance of CART, but poses unique mathematical challenges. It is perhaps not surprising then that, despite the widespread use of CART, there have been only a small handful of papers that study its theoretical

properties. For example, [15] study the asymptotic properties of CART in a fixed dimensional regime, en route to establishing consistency of Breiman's random forests for additive regression models. Another notable paper [8] provides oracle-type inequalities for the CART pruning algorithm proposed by [4], though the theory does not imply guarantees for out-out-sample prediction. What the existing literature currently lacks, however, is a more fine-grained analysis that reveals the unique advantages of tree learning with CART over other unstructured regression procedures, like vanilla $k$-NN or other kernel based estimators. Filling this theoretical gap, our main message is that, in certain settings, CART can identify low dimensional, latent structure in the data and adapt accordingly. We illustrate the adaptive properties of CART when the model is *sparse*, namely, when the output depends only on a small, unknown subset of the input variables—thereby circumventing the curse of dimensionality.

Arguably the most difficult technical aspect of studying decision trees (and for that matter, any adaptive partitioning-based predictor) is understanding their approximation error, or at least pinning down conditions on the data that enable such an endeavor. Indeed, most existing convergence results for decision trees or ensembles thereof bound the expected (over the training data) prediction error by the size (i.e., the diameter) of the terminal nodes and show that they vanish with the depth of the tree, ensuring that the approximation error does so also [6, 16]. Others [15] control the variation of the regression function inside the terminal nodes, without explicitly controlling their diameters, though the effect is similar. While these techniques can be useful to prove consistency statements, they are not generally delicate enough to capture the adaptive properties of the tree. It also often requires making strong assumptions about the tree construction. To address this shortcoming, in contrast, we use the fact that the prediction error is with high-probability (over the training data) bounded by the training error plus a complexity term. One of our crucial insights is that we can avoid using the node diameters as a proxy for the approximation error and, instead, directly bound the training error in terms of data dependent quantities (like the Pearson correlation coefficient) that are more transparent and interpretable, thereby facilitating our analysis and allowing us to prove more fine-grained results.

## 1.1 Learning setting

Let us now describe the learning setting and framework that we will operate under for the rest of the paper. For clarity and ease of exposition, we focus specifically on *regression trees*, where the target outcome is a continuous real value. We assume the training data is $\mathcal{D}_n = \{(\mathbf{X}_1, Y_1), \ldots, (\mathbf{X}_n, Y_n)\}$, where $(\mathbf{X}_i, Y_i)$, $1 \le i \le n$ are i.i.d. with common joint distribution $\mathbb{P}_{\mathbf{X},Y}$. Here, $\mathbf{X}_i \in [0,1]^d$ is the input and $Y_i \in \mathbb{R}$ is a continuous response (or output) variable. A generic pair of variables will be denoted as $(\mathbf{X}, Y)$. A generic coordinate of $\mathbf{X}$ will be denoted by $X$, unless there is a need to highlight the dependence on the $j^{\text{th}}$ coordinate index, denoted by $X_j$, or additionally on the $i^{\text{th}}$ data point, denoted by $X_{ij}$. Using squared error loss $L(Y, Y') = (Y - Y')^2$ as the performance metric, our goal is to predict $Y$ at a new point $\mathbf{X} = \mathbf{x}$ via a tree structured prediction rule $\widehat{Y}(\mathbf{x}) = \widehat{Y}(\mathbf{x}; \mathcal{D}_n)$. The training error and mean squared prediction error are, respectively,

$$\overline{\text{err}}(\widehat{Y}) := \frac{1}{n} \sum_{i=1}^{n} (Y_i - \widehat{Y}(\mathbf{X}_i))^2 \quad \text{and} \quad \text{Err}(\widehat{Y}) := \mathbb{E}_{(\mathbf{X}',Y')}[(Y' - \widehat{Y}(\mathbf{X}'))^2],$$

where $(\mathbf{X}', Y')$ denotes an independent copy of $(\mathbf{X}, Y)$. For data $\{(\mathbf{X}_1, U_1, V_1), \ldots, (\mathbf{X}_n, U_n, V_n)\}$, we let

$$\widehat{\rho}(U, V \mid \mathbf{X} \in A) := \frac{\frac{1}{N} \sum_{\mathbf{X}_i \in A} (U_i - \overline{U})(V_i - \overline{V})}{\sqrt{\frac{1}{N} \sum_{\mathbf{X}_i \in A} (U_i - \overline{U})^2 \times \frac{1}{N} \sum_{\mathbf{X}_i \in A} (V_i - \overline{V})^2}},$$

($A$ is a subset, $N = \#\{\mathbf{X}_i \in A\}$, $\overline{U} = \frac{1}{N} \sum_{\mathbf{X}_i \in A} U_i$, and $\overline{V} = \frac{1}{N} \sum_{\mathbf{X}_i \in A} V_i$) denote the empirical Pearson product-moment correlation coefficient, given $\mathbf{X} \in A$, and let $\rho(U, V \mid \mathbf{X} \in A)$ be its infinite sample counterpart. If $U_i = g(X_{ij})$ for a univariate function $g(\cdot)$ of a coordinate $X_j$, we write $\widehat{\rho}(g(X_j), V \mid \mathbf{X} \in A)$ or $\rho(g(X_j), V \mid \mathbf{X} \in A)$. For brevity, we let $\widehat{\sigma}_Y^2$ denote the sample variance of the response values $Y_1, Y_2, \ldots, Y_n$ in the training data. The $r^{\text{th}}$ derivative of a real valued function $g(\cdot)$ is denoted by $g^{(r)}(\cdot)$. Proofs of all forthcoming results are given in the supplement.

## 2 Preliminaries

As mentioned earlier, regression trees are commonly constructed with Classification and Regression Tree (CART) [4] methodology. The primary objective of CART is to find partitions of the input variables that produce minimal variance of the response values (i.e., minimal sum of squares error

with respect to the average response values). Because of the computational infeasibility of choosing the best overall partition, CART decision trees are greedily grown with a procedure in which binary splits recursively partition the tree into near-homogeneous terminal nodes. An effective binary split partitions the data from the parent tree node into two daughter nodes so that the homogeneity in each of the daughter nodes, as measured through the *impurity*, is reduced.

The CART algorithm is comprised of two elements—a growing procedure and a pruning procedure. The growing procedure constructs from the data a maximal binary tree $T_{\max}$ by the recursive partitioning scheme; the pruning procedure selects, among all the subtrees of $T_{\max}$, a sequence of subtrees that greedily optimize a cost function.

## 2.1 Growing the tree

Let us now describe the tree construction algorithm with additional detail. Consider splitting a regression tree $T$ at a node $\mathrm{t}$. Let $s$ be a candidate split point for a generic variable $X$ that divides $\mathrm{t}$ into left and right daughter nodes $\mathrm{t}_L$ and $\mathrm{t}_R$ according to whether $X \leq s$ or $X > s$. These two nodes will be denoted by $\mathrm{t}_L = \{\mathbf{X} \in \mathrm{t} : X \leq s\}$ and $\mathrm{t}_R = \{\mathbf{X} \in \mathrm{t} : X > s\}$. As mentioned previously, a tree is grown by recursively reducing node impurity. Impurity for regression trees is determined by the within node sample variance

$$\widehat{\Delta}(\mathrm{t}) := \widehat{\mathrm{VAR}}(Y \mid \mathbf{X} \in \mathrm{t}) = \frac{1}{N(\mathrm{t})} \sum_{\mathbf{X}_i \in \mathrm{t}} (Y_i - \overline{Y}_\mathrm{t})^2, \tag{1}$$

where $\overline{Y}_\mathrm{t} = \frac{1}{N(\mathrm{t})} \sum_{\mathbf{X}_i \in \mathrm{t}} Y_i$ is the sample mean for $\mathrm{t}$ and $N(\mathrm{t}) = \#\{\mathbf{X}_i \in \mathrm{t}\}$ is the number of data points in $\mathrm{t}$. Similarly, the within node sample variance for a daughter node is

$$\widehat{\Delta}(\mathrm{t}_L) = \frac{1}{N(\mathrm{t}_L)} \sum_{\mathbf{X}_i \in \mathrm{t}_L} (Y_i - \overline{Y}_{\mathrm{t}_L})^2, \quad \widehat{\Delta}(\mathrm{t}_R) = \frac{1}{N(\mathrm{t}_R)} \sum_{\mathbf{X}_i \in \mathrm{t}_R} (Y_i - \overline{Y}_{\mathrm{t}_R})^2,$$

where $\overline{Y}_{\mathrm{t}_L}$ is the sample mean for $\mathrm{t}_L$ and $N(\mathrm{t}_L)$ is the sample size of $\mathrm{t}_L$ (similar definitions apply to $\mathrm{t}_R$). The parent node $\mathrm{t}$ is split into two daughter nodes using the variable and split point producing the largest decrease in impurity. For a candidate split $s$ for $X$, this decrease in impurity equals [4, Definition 8.13]

$$\widehat{\Delta}(s, \mathrm{t}) := \widehat{\Delta}(\mathrm{t}) - [\widehat{P}(\mathrm{t}_L)\widehat{\Delta}(\mathrm{t}_L) + \widehat{P}(\mathrm{t}_R)\widehat{\Delta}(\mathrm{t}_R)], \tag{2}$$

where $\widehat{P}(\mathrm{t}_L) = N(\mathrm{t}_L)/N(\mathrm{t})$ and $\widehat{P}(\mathrm{t}_R) = N(\mathrm{t}_R)/N(\mathrm{t})$ are the proportions of data points in $\mathrm{t}$ that are contained in $\mathrm{t}_L$ and $\mathrm{t}_R$, respectively.

The tree $T$ is grown recursively by finding the variable $\hat{j}$ and split point $\hat{s} = \hat{s}_{\hat{j}}$ that maximizes $\widehat{\Delta}(s, \mathrm{t})$. Note that for notational brevity, we suppress the dependence on the input coordinate index $j$. The output $\widehat{Y}(T)$ of the tree at a terminal node $\mathrm{t}$ is the least squares predictor, namely, $\widehat{Y}(T, \mathbf{x}) = \overline{Y}_\mathrm{t}$ for all $\mathbf{x} \in \mathrm{t}$.

## 2.2 Pruning the tree

The CART growing procedure stops once a maximal binary tree $T_{\max}$ is grown (i.e., when the terminal nodes contain at least a single data point). However, $\widehat{Y}(T_{\max})$ is generally not a good predictor, since it will tend to overfit the data and therefore generalize poorly to unseen data. This effect can be mitigated by complexity regularization. Removing portions of the overly complex tree (i.e., via pruning) is one way of reducing its complexity and improving performance. We will now describe such a procedure.

We say that $T$ is a pruned subtree of $T'$, written as $T \preceq T'$, if $T$ can be obtained from $T'$ by iteratively merging any number of its internal nodes. A pruned subtree of $T_{\max}$ is defined as any binary subtree of $T_{\max}$ having the same root node as $T_{\max}$. The number of terminal nodes in a tree $T$ is denoted $|T|$. Given a subtree $T$ and temperature $\alpha > 0$, we define the penalized cost function

$$R_\alpha(\widehat{Y}(T)) := \overline{\mathrm{err}}(\widehat{Y}(T)) + \alpha|T|. \tag{3}$$

As shown in [4, Section 10.2], the smallest minimizing subtree for the temperature $\alpha$,

$$\widehat{T} \in \underset{T \preceq T_{\max}}{\arg\min} R_\alpha(\widehat{Y}(T)),$$

exists and is unique (smallest in the sense that if $T' \in \arg\min_{T \preceq T_{\max}} R_\alpha(\widehat{Y}(T))$, then $\widehat{T} \preceq T'$). For a fixed $\alpha$, the optimal subtree $\widehat{T}$ can be found efficiently by weakest link pruning [4, 9], i.e., by successively collapsing the internal node that increases $\overline{\mathrm{err}}(\widehat{Y}(T))$ the least, until we arrive at the single-node tree consisting of the root node. Good values of $\alpha$ can be selected using cross-validation, for example, though analyzing the effect of such a procedure is outside the scope of the present paper.

Our first result shows that, with high probability, the test error of the pruned tree $\widehat{T}$ on new data is bounded by a multiple of $\min_{T \preceq T_{\max}} R_\alpha(\widehat{Y}(T))$ plus lower order terms.

**Theorem 1.** *Let $\widehat{T}$ be the smallest minimizer of ([3](#)). Suppose $Y = f(\mathbf{X})$, $B = \sup_{\mathbf{x}} |f(\mathbf{x})| < \infty$, $n > (d+1)/2$, and $\alpha > \frac{27B^2(d+1)\log(2en/(d+1))}{n}$. Then, with probability at least $1 - \delta$ over the training sample $\mathcal{D}_n$,*

$$Err(\widehat{Y}(\widehat{T})) \le 4 \min_{T \preceq T_{max}} R_\alpha(\widehat{Y}(T)) + \frac{54B^2 \log(2/\delta)}{n}.$$

Similar bounds hold for the excess risk for binary classification, i.e., $Y \in \{0,1\}$, since, in this case, the squared error impurity ([1](#)) equals one-half of the so-called *Gini* impurity used for classification trees (which output the majority vote in each terminal node). See also [14] for results of a similar flavor when the penalty is proportional to $\sqrt{|T|}$.

In what follows, we let $T_K \preceq T_{\max}$ denote a fully grown binary tree of depth $K = \Theta(\log_2(n))$, i.e., we stop splitting if (1) the node contains a single data point, (2) all response values in the node are the same, or (3) a depth of $K$ is reached, whichever occurs sooner. We also let $\widehat{T}$ be the smallest minimizer of the cost function ([3](#)) with temperature $\alpha = \Theta((d/n)\log(n/d))$.

## 3 Bounding the Training Error

In the previous section, Theorem [1](#) showed that, with high probability, the test error is bounded by a multiple of the cost function ([3](#)) at its minimum (plus lower order terms). Since the cost function is defined as the training error plus penalty term, the next step in our course of study is to understand how the training error of CART behaves.

### 3.1 Splitting criterion and Pearson correlation

Before we begin our analysis of the training error, we first digress back to the tree construction algorithm and give an alternative characterization of the objective. Now, the use of the sum of squares impurity criterion $\widehat{\Delta}(s, \mathrm{t})$ with averages in the terminal nodes permits further simplifications of the formula ([2](#)) above. For example, using the sum of squares decomposition, $\widehat{\Delta}(s, \mathrm{t})$ can equivalently be expressed as [4, Section 9.3]

$$\widehat{P}(\mathrm{t}_L)\widehat{P}(\mathrm{t}_R)\big(\overline{Y}_{\mathrm{t}_L} - \overline{Y}_{\mathrm{t}_R}\big)^2, \tag{4}$$

which is commonly used for its computational appeal—that is, one can find the best split for a continuous variable with just a single pass over the data, without the need to calculate multiple averages and sums of squared differences for these averages, as required with ([2](#)). Yet another way to view $\widehat{\Delta}(s, \mathrm{t})$, which does not appear to have been considered in past literature and will prove to be useful for our purposes, is via its equivalent representation as $\widehat{\Delta}(\mathrm{t}) \times \widehat{\rho}^2(\widetilde{Y}, Y \mid \mathbf{X} \in \mathrm{t})$, where

$$\widehat{\rho}(\widetilde{Y}, Y \mid \mathbf{X} \in \mathrm{t}) := \frac{\frac{1}{N(\mathrm{t})} \sum_{\mathbf{X}_i \in \mathrm{t}} (\widetilde{Y}_i - \overline{Y}_{\mathrm{t}})(Y_i - \overline{Y}_{\mathrm{t}})}{\sqrt{\frac{1}{N(\mathrm{t})} \sum_{\mathbf{X}_i \in \mathrm{t}} (\widetilde{Y}_i - \overline{Y}_{\mathrm{t}})^2 \times \frac{1}{N(\mathrm{t})} \sum_{\mathbf{X}_i \in \mathrm{t}} (Y_i - \overline{Y}_{\mathrm{t}})^2}} \ge 0 \tag{5}$$

is the Pearson product-moment correlation coefficient between the decision stump

$$\widetilde{Y} := \overline{Y}_{\mathrm{t}_L} \mathbf{1}(X \le s) + \overline{Y}_{\mathrm{t}_R} \mathbf{1}(X > s) \tag{6}$$

and response variable $Y$ within $\mathrm{t}$ (for the proof, see Lemma [A.1](#) in the supplement).[1] Hence, at each node, CART seeks the decision stump most correlated in magnitude with the response variable along

a particular variable, i.e.,

$$\hat{s} \in \arg\max_{s} \widehat{\Delta}(s, \mathrm{t}) = \arg\max_{s} \widehat{\rho}(\widetilde{Y}, Y \mid \mathbf{X} \in \mathrm{t}). \tag{7}$$

Equivalently, CART splits along variables with decision stumps that are most correlated with the residuals $Y_i - \overline{Y}_{\mathrm{t}}$ of the current fit $\overline{Y}_{\mathrm{t}}$. As with $r^2$ for simple linear regression, the squared correlation $\widehat{\rho}^2(\widetilde{Y}, Y \mid \mathbf{X} \in \mathrm{t})$ equals the *coefficient of determination* $R^2$, in the sense that it describes the fraction of the variance in $Y$ that is explained by a decision stump $\widetilde{Y}$ in $X$, since $\widehat{\rho}^2(\widetilde{Y}, Y \mid \mathbf{X} \in \mathrm{t}) = \widehat{\Delta}(s, \mathrm{t})/\widehat{\Delta}(\mathrm{t}) = 1 - \frac{1}{N(\mathrm{t})} \sum_{\mathbf{X} \in \mathrm{t}} (Y_i - \widetilde{Y}_i)^2 / \frac{1}{N(\mathrm{t})} \sum_{\mathbf{X} \in \mathrm{t}} (Y_i - \overline{Y}_{\mathrm{t}})^2$.

**Definition 1.** *We let $\widehat{Y}$ denote a decision stump $\widetilde{Y}$ with an optimal direction $\hat{j} \in \arg\max_{j=1,2,\ldots,d} \widehat{\Delta}(\hat{s}, \mathrm{t})$ and corresponding optimal split $\hat{s}$.*

We now introduce a data dependent quantity that will play a central role in determining the rates of convergence of the prediction error. For a univariate function class $\mathcal{G}$, we let $\widehat{\rho}_{\mathcal{G}}$ be the largest Pearson correlation between the response data $Y$ and a function in $\mathcal{G}$ of a single input coordinate for a worst-case node, i.e.,

$$\widehat{\rho}_{\mathcal{G}} := \min_{\mathrm{t}} \sup_{g(\cdot) \in \mathcal{G}, \, j=1,2,\ldots,d} |\widehat{\rho}(g(X_j), Y \mid \mathbf{X} \in \mathrm{t})|, \tag{8}$$

where the minimum runs over all internal nodes $\mathrm{t}$ in $T_K$. We will specifically focus on classes $\mathcal{G}$ that consist of decision stumps, and more generally, monotone functions.

### 3.2 Location of splits and Pearson correlation

Having already revealed the intimate role the correlation between the decision stump and response values (5) plays in the tree construction, it is instructive to explore this relationship with the location of the splits. In order to study this cleanly, let us for the moment work in an asymptotic data setting to determine the coordinates to split and their split points, i.e.,

$$\widehat{\Delta}(s, \mathrm{t}) \xrightarrow[n \to \infty]{} \Delta(s, \mathrm{t}) := \Delta(\mathrm{t}) - [P(\mathrm{t}_L)\Delta(\mathrm{t}_L) + P(\mathrm{t}_R)\Delta(\mathrm{t}_R)], \tag{9}$$

where quantities without hats are the population level counterparts of the empirical quantities defined previously in (2). A decision stump (6) with an optimal theoretical direction $j^*$ and corresponding optimal theoretical split $s^* = s^*_{j^*}$ is denoted by $\widehat{Y}^*$. Now, if the number of data points within $\mathrm{t}$ is large and $\Delta(s, \mathrm{t})$ has a unique global maximum, then we can expect $\hat{s} \approx s^*$ (via an empirical process argument) and hence the infinite sample setting is a good approximation to CART with empirical splits, giving us some insights into its dynamics. Indeed, if $s^*$ is unique, [10, Theorem 2] shows that $\hat{s}$ converges in probability to $s^*$. With additional assumptions, one can go even further and characterize the rate of convergence. For example, [5, Section 3.4.2] and [1] provide cube root asymptotics for $\hat{s}$, i.e., $n^{1/3}(\hat{s} - s^*)$ converges in distribution.

Each node $\mathrm{t}$ is a Cartesian product of intervals. As such, the interval along variable $X$ in $\mathrm{t}$ is denoted by $[a, b]$, where $a < b$. The next theorem characterizes the relationship between an optimal theoretical split $s^*$ and infinite sample correlation $\rho(\widehat{Y}, Y \mid \mathbf{X} \in \mathrm{t}) \stackrel{\text{a.s.}}{:=} \lim_n \widehat{\rho}(\widehat{Y}^*, Y \mid \mathbf{X} \in \mathrm{t})$ for a deterministic node $\mathrm{t}$ (the limit exists by the uniform law of large numbers). The proof is based on the first-order optimality condition, namely, $\frac{\partial}{\partial s}\Delta(s, \mathrm{t})\mid_{s=s^*} = 0$.

**Theorem 2.** *Suppose $\mathbf{X}$ is uniformly distributed on $[0, 1]^d$ and $\Delta(s^*, \mathrm{t}) > 0$. For a deterministic parent node $\mathrm{t}$, an optimal theoretical split $s^* \in [a, b]$ along variable $X$ has the form*

$$\frac{a + b}{2} \pm \frac{b - a}{2}\sqrt{\frac{v}{v + \rho^2(\widehat{Y}^*, Y \mid \mathbf{X} \in \mathrm{t})}}, \tag{10}$$

*where $v = \frac{(\mathbb{E}[Y \mid \mathbf{X} \in \mathrm{t}, \, X = s^*] - \mathbb{E}[Y \mid \mathbf{X} \in \mathrm{t}])^2}{VAR(Y \mid \mathbf{X} \in \mathrm{t})}$.*

Expression (10) in Theorem 2 reveals that an optimal theoretical split $s^*$ is a perturbation of the median $(a + b)/2$ of the conditional distribution $X \mid \mathbf{X} \in \mathrm{t}$, where the gap is governed by the correlation $\rho(\widehat{Y}^*, Y \mid \mathbf{X} \in \mathrm{t})$. These correlations control the local and directional granularity of the partition of the input domain. Splits along input coordinates that contain a strong signal, i.e.,

$\rho(\widehat{Y}^*, Y \mid \mathbf{X} \in \mathrm{t}) \gg 0$, tend to be further away from the parent node edges, thereby producing side lengths $[a, b]$ that are on average narrower. At the other extreme, the correlation is weakest when there is no signal in the splitting direction or when the response values in the node are not fit well by a decision stump—yielding either $s^* \approx a + (b-a)\rho^2(\widehat{Y}^*, Y \mid \mathbf{X} \in \mathrm{t})/(4v)$ or $s^* \approx b - (b-a)\rho^2(\widehat{Y}^*, Y \mid \mathbf{X} \in \mathrm{t})/(4v)$—and hence the predicted output in one of the daughter nodes does not change by much. For example, if $Y = g(X)$ is a sinusoidal waveform with large frequency $w$ (not fit well by a single decision stump) and $\mathrm{t}$ is the root node $[0, 1]^d$, then $v = \Theta(1)$ and $\rho(\widehat{Y}^*, Y \mid \mathbf{X} \in \mathrm{t}) = \Theta(1/\sqrt{w})$, and hence by (10), either $s^* = \Theta(1/w)$ or $s^* = 1 - \Theta(1/w)$ (see Lemma A.2 in the supplement). This phenomenon, where optimal splits concentrate at the endpoints of the node along noisy directions, has been dubbed 'end-cut preference' in the literature and has been known empirically since the inception of CART [10], [4, Section 11.8]. The theory above is also consistent with empirical studies on the local adaptivity of Breiman's random forests which use CART [12, Section 4].

### 3.3 Training error and Pearson correlation

In addition to determining the location of the splits, the correlation is also directly connected to the training error. Intuitively, the training error should small when CART finds decision stumps that have strong correlation with the response values in each node. More precisely, the following lemma reveals the importance of the correlation (5) in controlling the training error. It shows that each time a node $\mathrm{t}$ is split, the training error in $\mathrm{t}$ is reduced by a constant factor, namely, $\exp(-\widehat{\rho}^2(\widehat{Y}, Y \mid \mathbf{X} \in \mathrm{t}))$ or, uniformly, by $\exp(-\widehat{\rho}_{\mathcal{H}}^2)$, where $\mathcal{H}$ is the collection of all decision stumps and $\widehat{\rho}_{\mathcal{H}}$ is the quantity defined in (8). Recursing this contraction inequality over nodes at each level of the tree leads to the conclusion that the training error should be exponentially small in the depth $K$, provided the correlation at each node is large.

**Lemma 1.** *Almost surely,*

$$\frac{1}{N(\mathrm{t})} \sum_{\mathbf{X}_i \in \mathrm{t}} (Y_i - \widehat{Y}_i)^2 \leq \frac{1}{N(\mathrm{t})} \sum_{\mathbf{X}_i \in \mathrm{t}} (Y_i - \overline{Y}_\mathrm{t})^2 \times \exp(-\widehat{\rho}^2(\widehat{Y}, Y \mid \mathbf{X} \in \mathrm{t})), \qquad (11)$$

*and hence*

$$\overline{err}(\widehat{Y}(T_K)) \leq \widehat{\sigma}_Y^2 \exp(-K \times \widehat{\rho}_{\mathcal{H}}^2), \qquad (12)$$

*where $\mathcal{H}$ is the collection of all decision stumps.*

### 3.4 Size of Pearson correlation via comparison inequalities

Due to the importance of the correlation in controlling the training error, it is natural to ask when it will be large. We accomplish this by studying its size relative to the correlation between the data and another more flexible model. That is, we fit an arbitrary univariate function $g(X)$ of a generic coordinate $X$ to the data in the node and ask how large $\widehat{\rho}(\widehat{Y}, Y \mid \mathbf{X} \in \mathrm{t})$ is relative to $|\widehat{\rho}(g(X), Y \mid \mathbf{X} \in \mathrm{t})|$. Such a relationship will enable us to conclude that if $Y$ is locally correlated with $g(X)$ in the node, then so will $Y$ with an optimal decision stump $\widehat{Y}$. Before we continue, let us mention that studying $\widehat{\rho}(\widehat{Y}, Y \mid \mathbf{X} \in \mathrm{t})$ directly is hopeless since it not at all straightforward to disentangle the dependence on the data. Even if this could be done and a target population level quantity could be identified, it is difficult to rely on concentration of measure when $\mathrm{t}$ contains very few data points; a likely situation among deep nodes. Nevertheless, by definition of $\widehat{Y}$ via (7), we can construct a prior $\Pi(j, s)$ on coordinates $j$ and splits $s$, and lower bound $\widehat{\rho}(\widehat{Y}, Y \mid \mathbf{X} \in \mathrm{t})$ by

$$\int \widehat{\rho}(\widetilde{Y}, Y \mid \mathbf{X} \in \mathrm{t}) d\Pi(j, s), \qquad (13)$$

which is much less burdensome to analyze. Importantly, the prior can involve unknown quantities from the distribution of $(\mathbf{X}, Y)$. For a special choice of prior $\Pi$, (13) can be further lower bounded by

$$\widehat{\rho}(\widehat{Y}, Y \mid \mathbf{X} \in \mathrm{t}) \geq \text{constant} \times |\widehat{\rho}(g(X), Y \mid \mathbf{X} \in \mathrm{t})|. \qquad (14)$$

The constant in (14) depends on $g(\cdot)$, though importantly it is invariant to the scale of $g(\cdot)$. If $g(\cdot)$ belongs to a univariate model class $\mathcal{G}$, this constant can either be studied directly for the specific $g(\cdot)$ or minimized over $g(\cdot) \in \mathcal{G}$ to yield a more insightful lower bound. For certain model classes $\mathcal{G}$, the minimization problem turns out to be equivalent to a quadratic program, and the solution can be

obtained explicitly and used to prove the next set of results. Our first application of this technique shows that, despite fitting the data to a decision stump with *one* degree of freedom (i.e., the location of the split), CART behaves almost as if it fit the data to a monotone function with $N(\mathfrak{t}) - 1$ degrees of freedom, at the expense of a sublogarithmic factor in $N(\mathfrak{t})$. For example, the correlation between the response variable and the decision stump is, up to a sub-logarithmic factor, at least as strong as the correlation between the response variable and a linear or isotonic fit.

**Fact 1.** *Almost surely, uniformly over all monotone functions $g(\cdot)$ of $X$ in the node, we have*

$$\widehat{\rho}(\widehat{Y}, Y \mid \mathbf{X} \in \mathfrak{t}) \geq \frac{1}{\sqrt{1 + \log(2N(\mathfrak{t}))}} \times |\widehat{\rho}(g(X), Y \mid \mathbf{X} \in \mathfrak{t})|. \tag{15}$$

The previous fact also suggests that CART is quite good at fitting response values that have a local, low-dimensional, monotone relationship with the input variables. Note that because correlation is merely a measure of linear association, $|\widehat{\rho}(g(X), Y \mid \mathbf{X} \in \mathfrak{t})|$ can still be large for some monotone $g(\cdot)$, even if $Y$ is not approximately monotone in one coordinate. That is, $Y$ need only be locally correlated with such a function. On the other hand, if $Y$ has no signal in $X$, then [11, Lemma 1, Supplement] shows that, with high probability, $\widehat{\rho}(\widehat{Y}, Y \mid \mathbf{X} \in \mathfrak{t})$ is $\mathcal{O}(\sqrt{(\log N(\mathfrak{t}))/N(\mathfrak{t})})$.

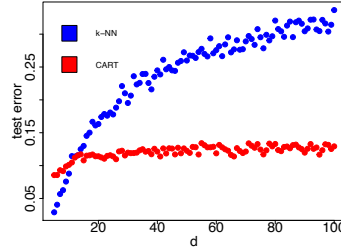

**(a)** Synthetic data. Prediction error of CART vs. $k$-NN as $d$ varies.

# 4 Main Results

In this section, we use the training error bound (12) and the device (13) for obtaining correlation comparison inequalities (à la Fact 1) to give bounds on the prediction error of CART. We first outline the high-level strategy. By Theorem 1, with high probability, the leading behavior of the prediction error $\mathrm{Err}(\widehat{Y}(\widehat{T}))$ is governed by $\inf_{T \preceq T_{\max}} R_\alpha(\widehat{Y}(T))$, which is smaller than the minimum of $R_\alpha(\widehat{Y}(T_K)) = \overline{\mathrm{err}}(\widehat{Y}(T_K)) + \alpha|T_K|$ over all fully grown trees $T_K$ of depth $K$ with $|T_K| \leq 2^K$, i.e.,

$$\inf_{K \geq 1} \{\overline{\mathrm{err}}(\widehat{Y}(T_K)) + \alpha 2^K\}. \tag{16}$$

Coupled with an informative bound on $\overline{\mathrm{err}}(\widehat{Y}(T_K))$, (16) can then be further bounded and solved. The proofs reveal that a good balance between the tree size and its goodness of fit occurs when $K$ is logarithmic in the sample size.

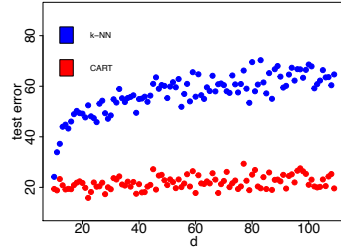

**(b)** Boston housing data. Prediction error of CART vs. $k$-NN as $d$ varies.

## 4.1 Asymptotic consistency rates for sparse additive models

Applying the training error bound (12) to (16) with $K = \lceil (\widehat{\rho}_{\mathcal{H}}^2 + \log 2)^{-1} \log(\widehat{\sigma}_Y^2/\alpha) \rceil$, we have from Theorem 1 that with probability at least $1 - \delta$,

$$\mathrm{Err}(\widehat{Y}(\widehat{T})) = \mathcal{O}\left(\widehat{\sigma}_Y^2 \left(\frac{d \log(n/d)}{n \widehat{\sigma}_Y^2}\right)^{\frac{\widehat{\rho}_{\mathcal{H}}^2}{\widehat{\rho}_{\mathcal{H}}^2 + \log 2}} + \frac{\log(1/\delta)}{n}\right). \tag{17}$$

It turns out that if $\mathbf{X}$ is uniformly distributed and $Y$ is a sparse additive model with $d_0$ component functions $g_j(\cdot)$, then $\widehat{\rho}_{\mathcal{H}}^2$ is asymptotically lower bounded by a constant multiple of $1/d_0$.

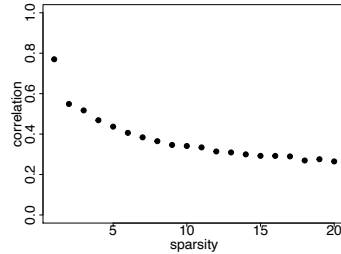

**(c)** Minimum correlation $\widehat{\rho}_{\mathcal{H}}$ (averaged over 10 independent replications) of all nodes of pruned CART tree as sparsity $d_0$ varies.

Thus, we find from (17) that if $d_0$ is fixed, then $\lim_n \mathrm{Err}(\widehat{Y}(\widehat{T})) \stackrel{\text{a.s.}}{=} 0$ even when the ambient dimension grows as $d = o(n)$. Note that such a statement is not possible for vanilla $k$-NN or other kernel based regression methods with nonadaptive weights, unless feature selection is performed beforehand. In fact, we show next that the prediction error rate that CART achieves is the same as what would be achieved by a standard kernel predictor *if* one had a priori knowledge of the locations of the $d_0$ relevant input variables that determine the output. A routine computer experiment

on synthetic or real data easily confirms this theory. In Fig. 1a and Fig. 1c, we generate 1000 samples from the model $Y = \sum_{j=1}^{d_0} g_j(X_j)$, where each $g_j(X_j)$ equals $\pm X_j^2$ (alternating signs) and $\mathbf{X} \sim \text{Uniform}([0,1]^d)$. In Fig. 1a, we plot the test error, averaged over 10 independent replications, of pruned CART vs. $k$-NN (with cross-validated $k$) as $d$ ranges from 5 to 100 with $d_0 = 5$ fixed. A similar experiment is performed in Fig. 1b on the Boston housing dataset [4, Section 8.2] ($d_0 = 10$ and $n = 506$), where we scale the inputs to be in $[0,1]$ and add $d - d_0$ noisy $\text{Uniform}([0,1])$ input variables. According to Theorem 3, the convergence rate of CART depends primarily on the sparsity $d_0$ and therefore its performance should not be adversely affected by growing $d$. Consistent with our theory, the prediction error of CART remains stable as $d$ increases, whereas $k$-NN does not adapt to the sparsity. Furthermore, Fig. 1c illustrates how $\widehat{\rho}_{\mathcal{H}}^2$ decays with $d_0$ if $d = 20$, thus corroborating with the aforementioned asymptotic behavior of $\Omega(1/d_0)$.

**Theorem 3.** *Suppose $\mathbf{X}$ is uniformly distributed on $[0,1]^d$ and $Y = \sum_j g_j(X_j)$ is a sparse additive model with $d_0 \ll d$ smooth component functions $g_j(\cdot)$, where each function is not too locally 'flat' in the sense that*

$$\sup_x \inf\{r \geq 1 : g_j^{(r)}(\cdot) \text{ exists, continuous, and nonzero at } x\} < \infty. \tag{18}$$

*Then there exists a constant $C > 0$ that is independent of $d_0$ such that, almost surely, $\liminf_n \widehat{\rho}_{\mathcal{H}}^2 \geq C/d_0$, and*

$$\limsup_n \frac{Err(\widehat{Y}(\widehat{T}))}{((d/n)\log(n/d))^{\Omega(1/d_0)}} \overset{a.s.}{=} \mathcal{O}(1). \tag{19}$$

**Remark 1.** *For independent, continuous marginal input variables $X_j$, there is no loss of generality in assuming uniform distributions in Theorem 3. Indeed, CART decision trees are invariant to strictly monotone transformations of $X_j$. One such transformation is the marginal cumulative distribution function $F_{X_j}(\cdot)$, for which $F_{X_j}(X_j) \sim \text{Uniform}([0,1])$—and so the problem can immediately be reduced to the uniform case.*

Any nonconstant component function $g_j(\cdot)$ that admits a power series representation satisfies the hypothesis of Theorem 3, though, in general, the condition (18) accommodates functions that are not analytic or infinitely differentiable. In fact, even differentiability is not necessary—similar results hold if the $g_j(\cdot)$ are *step functions*, as we now show. To this end, assume that $Y = \sum_j g_j(X_j)$ is an additive model, where each component function $g_j(\cdot)$ is a bounded step function and the total number of constant pieces of $Y$ is $V$. We show in the supplement that each optimal split $\hat{s}$ in a node t satisfies

$$\max\{X_{ij} \in I : \mathbf{X}_i \in \text{t}\} \leq \hat{s} \leq \min\{X_{ij} \in I' : \mathbf{X}_i \in \text{t}\}, \tag{20}$$

for some direction $j$ and successive intervals $I$ and $I'$ on which $g_j(\cdot)$ is constant. For example, if $Y = c_1 \mathbf{1}(X_1 < s_1) + c_2 \mathbf{1}(X_2 < s_2)$, then the first split separates the data along $X_1$ at $s_1$ (resp. $X_2$ at $s_2$), and at the next level down, CART separates the data in both daughter nodes along $X_2$ at $s_2$ (resp. $X_1$ at $s_1$). Thus, in general, each empirical split always perfectly separates the data in the node between two adjacent constant values of a component function. A CART decision tree grown in this way will achieve zero training error once it has at least $V$ terminal nodes, i.e., $|T| \geq V$. This is in fact the same training error that would result from the global least squares projection onto the space of all step functions with $V$ constant pieces. From Theorem 1, we immediately obtain the following performance bound, which is the optimal $\mathcal{O}(1/n)$ parametric rate for prediction, up to a logarithmic factor in the sample size [17]. Notice that we do not make any assumptions on the input distribution.

**Theorem 4.** *Suppose $Y = \sum_j g_j(X_j)$, where each component function $g_j(\cdot)$ is a bounded step function and the total number of constant pieces of $Y$ is $V$. With probability at least $1 - \delta$,*

$$Err(\widehat{Y}(\widehat{T})) = \mathcal{O}\left(\frac{Vd\log(n/d)}{n} + \frac{\log(1/\delta)}{n}\right). \tag{21}$$

## 4.2 Finite sample consistency rates for general sparse models

Using Fact 1, we now provide results of a similar flavor for more general regression models under a mild assumption on the largest number of data points in a node at level $k$ in $T_K$, denoted by $N_k$. Importantly, our theory only requires that each $N(\text{t})$ is *upper* bounded at every level of the tree. This condition still allows for nodes that have very few data points, which is typical for trees trained in practice. Contrast this assumption with past work on tree learning (including tree ensembles like random forests) that requires each $N(\text{t})$ to be *lower* bounded [4, Section 12.2], [13, 16, 6].

**Assumption 1.** *For some constants $a \geq 0$ and $A > 0$, the largest number of data points in a node at level $k$ in $T_K$ satisfies $N_k \leq Ank^a/2^k$, for $k = 1, 2, \ldots, K = \Theta(\log_2(n))$.*

Recall the quantity $\widehat{\rho}_{\mathcal{G}}$ defined in (8), namely, the largest correlation between the response data $Y$ and a function in $\mathcal{G}$ for a worst-case node. Our next theorem shows that if $\mathcal{M}$ is the collection of all monotone (i.e., increasing or decreasing) functions, then

$$\widehat{\rho}_{\mathcal{M}} = \min_{\text{t}} \sup_{g(\cdot) \text{ monotone}, \, j=1,2,\ldots,d} |\widehat{\rho}\left(g(X_j), Y \mid \mathbf{X} \in \text{t}\right)|$$

governs the rate at which the training error and prediction error decrease. Both errors are small if the local monotone dependence between $\mathbf{X}$ and $Y$ is high; that is, if CART partitions the input domain into pieces where the response variable is locally monotone in a few of the input coordinates.

**Theorem 5.** *Let $Y = f(\mathbf{X})$, where $f(\cdot)$ is a bounded function. Under Assumption 1, almost surely,*

$$\overline{err}(\widehat{Y}(T_K)) \leq \widehat{\sigma}_Y^2 \left(1 - \frac{K}{\log_2(4K^a An)}\right)^{\widehat{\rho}_{\mathcal{M}}^2}. \tag{22}$$

*Furthermore, with probability at least $1 - \delta$,*

$$Err(\widehat{Y}(\widehat{T})) = \mathcal{O}\left(\widehat{\sigma}_Y^2 \left(\frac{\log((d/\widehat{\sigma}_Y^2)\log^{2+a}(n))}{\log(n)}\right)^{\widehat{\rho}_{\mathcal{M}}^2} + \frac{\log(1/\delta)}{n}\right). \tag{23}$$

We will now argue that $\widehat{\rho}_{\mathcal{M}}$ is an empirical measure of the local dimensionality of $Y$. More specifically, we argue that if CART effectively partitions the input domain so that, in each node, $Y$ is locally correlated with sparse additive models with $d_0 \ll d$ monotone component functions, then $\widehat{\rho}_{\mathcal{M}}^2 = \Omega(1/d_0)$. To see why this assertion is true, suppose $g_1(X_1), g_2(X_2), \ldots, g_d(X_d)$ is an arbitrary collection of $d$ univariate functions from $\mathcal{M}$. However, suppose that only a subset of $d_0$ of the input variables $X_1, X_2, \ldots, X_d$ locally affect $Y$ in each node. Then, it can be shown (see Lemma A.3 in the supplement) that there is some node t and sparse additive model $Y_0$ with $d_0$ component functions of the form $\pm g_j(X_j)$, corresponding to the $d_0$ input variables that locally affect $Y$, such that

$$\widehat{\rho}_{\mathcal{M}}^2 \geq \frac{\widehat{\rho}^2(Y_0, Y \mid \mathbf{X} \in \text{t})}{d_0} = \Omega(1/d_0), \tag{24}$$

almost surely. The above statement is reminiscent of Theorem 3 in which $\widehat{\rho}_{\mathcal{H}}^2 = \Omega(1/d_0)$ controls the convergence rate of the prediction error when the underlying regression model is additive. Though, in contrast, note that (24) holds regardless of the dependence structure between the $d_0$ input coordinates that matter and the $d - d_0$ input coordinates that do not. Thus, (24) and Theorem 5 together suggest that it is possible to achieve rates of the form $(\log(d)/\log(n))^{\Omega(1/d_0)}$ in fairly general settings.

## 5 Extensions to Tree Ensembles

Key to our analysis of CART was the ability to connect the training error to the objective function of the growing procedure, as in Lemma 1. Establishing similar relationships is not as easy with trees that are constructed from bootstrap samples or random subsets of input variables. Nevertheless, we mention a few ideas for future work. Suppose $\widehat{Y}(\mathcal{T}) = (1/m)\sum_T \widehat{Y}(T)$ is the output of an ensemble $\mathcal{T}$ of $m$ decision trees from a random forest. By convexity of the squared error loss [3, Section 11] or [2, Section 4.1], we have $Err(\widehat{Y}(\mathcal{T})) \leq (1/m)\sum_T Err(\widehat{Y}(T))$, where the prediction error is averaged with respect to the tree randomization mechanism. Using the law of total expectation by conditioning on each realization of the (random) tree, $Err(\widehat{Y}(T))$ can be decomposed into quantities that involve the prediction error of a fixed (non-random) tree, for which our previous results can be applied. We will leave the exact details of these extensions for future work.

## 6 Conclusion

A key strength of CART decision trees is that they can exploit local, low dimensionality of the model—via a built-in, automatic dimension reduction mechanism. This is particularly useful since many real-world input/output systems are locally approximated by simple model forms with only a few variables. Adaptivity with CART is made possible by the recursive partitioning of the input space, in which optimal splits are increasingly affected by local qualities of the data as the tree is grown. To illustrate this ability, we identified settings where CART adapts to the unknown sparsity of the model. To the best of our knowledge, the consistency rates given here are the first of their kind for CART decision trees.

## Broader Impact

**Who may benefit from this research.** There are at least two groups of people who will benefit—either directly or indirectly—from this research.

1. Decision makers across a variety of domains, especially those with limited training in statistics. CART has enabled data-driven decision making in multiple high-stakes domains (e.g., business, medicine, and policy) over the past three decades. In particular, those who do not have a formal quantitative background will benefit from the intuitive and interpretable nature of CART and its quick and easy implementation.

2. Members of the society who may be have faced ethical/fairness concerns associated with their data and its use. As this paper has demonstrated, CART forms predictions by emphasizing variables that are more relevant to the output. In a social science context, this suggests that CART may focus more on key diagnostic information (e.g., education, income) without being influenced by potentially non-diagnostic variables that other methods may have falsely deemed relevant (e.g., gender, race).

**Who may be put at a disadvantage from this research.** There is no foreseeable population who may be put at a disadvantage from this research.

**What are the consequences of failure of the system.** Overreliance on any prediction method can have obvious, negative real-world consequences, particularly when the prediction method is prone to failure. CART suffers from a couple of pitfalls: instability (i.e., small perturbations in the training samples may significantly change the structure of an optimal tree and consequent predictions) and difficulty in accurately approximating certain simple models, such as linear or, more generally, additive, if given insufficient or low quality data.

**Whether the task/method leverages biases in the data.** While CART is not impervious to all pre-existing biases in the data (e.g., those arising from systematic measurement errors at the data collection stage), as we have shown, it is less susceptible to the presence of additional, non-diagnostic variables in the data. Consequently, CART has the potential to mitigate the negative consequences of biasing information that is inevitable with most datasets.

## Acknowledgments and Disclosure of Funding

The author is indebted to Min Xu, Minge Xie, Samory Kpotufe, Robert McCulloch, Andy Liaw, Richard Baumgartner, Regina Liu, Cun-Hui Zhang, Michael Kosorok, Jianqing Fan, and Matias Cattaneo for helpful discussions and feedback. This research was supported in part by NSF DMS #1915932 and NSF TRIPODS DATA-INSPIRE CCF #1934924.

## Footnotes

[1] It should be stressed that the alternative characterization of the splitting criterion ([2](#)) in terms of a correlation is unique to the squared error impurity with (constant) averages in the terminal nodes.

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
