[Supplementary Material]

# Supplementary Material

In Appendix A, we provide proofs of Theorem 1, Theorem 2, Lemma 1, Fact 1, Theorem 3, Theorem 4, and Theorem 5 from the main body of the paper. We also state and prove any supporting lemmas in Appendix B.

As a general rule, if the coordinate index $j$ is omitted on any quantity that should otherwise depend on $j$, it should be understood that we are considering a generic variable $X$. Similar conventions apply to an optimal empirical and theoretical split coordinate index, $\hat{j}$ and $j^*$, respectively.

## A  Main Proofs

**Lemma A.1** (Equivalence between the decrease in impurity and Pearson correlation from Section 3.1)**.**

$$\widehat{\rho}(\widetilde{Y}, Y \mid \mathbf{X} \in t) = \sqrt{\widehat{\Delta}(s, t)/\widehat{\Delta}(t)} \geq 0.$$

*Proof.* By expanding the sum of squares in (2), it can easily be shown that $\widehat{\Delta}(s, t)$ equals

$$\widehat{P}(t_L)(\overline{Y}_{t_L})^2 + \widehat{P}(t_R)(\overline{Y}_{t_R})^2 - (\overline{Y}_t)^2,$$

which is further equal to both $\frac{1}{N(t)}\sum_{\mathbf{X}_i \in t}(\widetilde{Y}_i - \overline{Y}_t)^2$ and $\frac{1}{N(t)}\sum_{\mathbf{X}_i \in t}(\widetilde{Y}_i - \overline{Y}_t)(Y_i - \overline{Y}_t)$. Thus,

$$
\begin{aligned}
\widehat{\rho}(\widetilde{Y}, Y \mid \mathbf{X} \in t) &= \frac{\frac{1}{N(t)}\sum_{\mathbf{X}_i \in t}(\widetilde{Y}_i - \overline{Y}_t)(Y_i - \overline{Y}_t)}{\sqrt{\frac{1}{N(t)}\sum_{\mathbf{X}_i \in t}(\widetilde{Y}_i - \overline{Y}_t)^2 \times \frac{1}{N(t)}\sum_{\mathbf{X}_i \in t}(Y_i - \overline{Y}_t)^2}} \qquad (\text{A.1})\\
&= \frac{\widehat{P}(t_L)(\overline{Y}_{t_L})^2 + \widehat{P}(t_R)(\overline{Y}_{t_R})^2 - (\overline{Y}_t)^2}{\sqrt{(\widehat{P}(t_L)(\overline{Y}_{t_L})^2 + \widehat{P}(t_R)(\overline{Y}_{t_R})^2 - (\overline{Y}_t)) \times \widehat{\Delta}(t)}}\\
&= \sqrt{\frac{\widehat{P}(t_L)(\overline{Y}_{t_L})^2 + \widehat{P}(t_R)(\overline{Y}_{t_R})^2 - (\overline{Y}_t)^2}{\widehat{\Delta}(t)}}\\
&= \sqrt{\widehat{\Delta}(s, t)/\widehat{\Delta}(t)}.
\end{aligned}
$$

Note that the mean of the decision stump $\widetilde{Y}$ in t is in fact $\overline{Y}_t$, which is why it appears in the formula (A.1) for the Pearson correlation. $\square$

**Lemma A.2** (Example from Section 3.2)**.** *Let $Y = \sin(2\pi w X)$ for some positive integer $w$ and $t = [0, 1]^d$. Then,*

$$\rho(\widehat{Y}^*, Y \mid \mathbf{X} \in t) = \Theta(1/\sqrt{w}), \ s^* = \Theta(1/w), \ and \ s^* = 1 - \Theta(1/w).$$

*Proof.* Elementary calculations reveal that $\Delta(s, t) = \frac{(1 - \cos(2\pi ws))^2}{4\pi^2 w^2 s(1-s)} = \frac{(1 - \cos(2\pi w(1-s)))^2}{4\pi^2 w^2 s(1-s)}$. It can be seen from this expression that the maximizers satisfy $s^* = \Theta(1/w)$ and $s^* = 1 - \Theta(1/w)$ and thus $\Delta(s^*, t) = \Theta(1/w)$. Since $\Delta(t) = 1/2$, we have from the infinite sample analog of Lemma A.1 that $\rho(\widehat{Y}^*, Y \mid \mathbf{X} \in t) = \sqrt{\Delta(s^*, t)/\Delta(t)} = \Theta(1/\sqrt{w})$. $\square$

**Lemma A.3** (Inequality (24) from Section 4.2)**.** *Let $g_1(X_1), g_2(X_2), \ldots, g_d(X_d)$ be univariate functions and let $Y_0 = \sum_j w_j g_j(X_j)$ consist of a subset of $d_0$ component functions $g_j(\cdot)$, where $w_j \in \{-1, +1\}$, and $\mathbf{w} = (w_j)_j$. Then,*

$$\max_{j=1,2,\ldots,d} \widehat{\rho}^2(g_j(X_j), Y \mid \mathbf{X} \in t) \geq \frac{\min_{\mathbf{w}} \widehat{\rho}^2(Y_0, Y \mid \mathbf{X} \in t)}{d_0}. \qquad (\text{A.2})$$

*Furthermore, if each $g_j(\cdot)$ has nonnegative Pearson correlation with the others in the node, then*

$$\max_{j=1,2,\ldots,d} \widehat{\rho}^2(g_j(X_j), Y \mid \mathbf{X} \in t) \geq \frac{\widehat{\rho}^2(Y_0, Y \mid \mathbf{X} \in t)}{d_0}, \qquad (\text{A.3})$$

*where $Y_0 = \sum_j g_j(X_j)$.*

*Proof.* Before we proceed with proving the lemma, we first establish some shorthand notation. Let $\widehat{\sigma}_h^2(\mathsf{t})$ denote the empirical variance of a function $h(\mathbf{X})$ in $\mathsf{t}$, i.e., $\widehat{\sigma}_h^2(\mathsf{t}) = \widehat{\mathrm{VAR}}(h(\mathbf{X}) \mid \mathbf{X} \in \mathsf{t})$. Define the discrete prior $\pi(j, \mathbf{w})$ on the component function index $j$ and sign vector $\mathbf{w}$ of $Y_0$ by

$$\pi(j, \mathbf{w}) = \frac{\widehat{\sigma}_{w_j g_j}(\mathsf{t})}{2^{d_0} \sum_{j'} \widehat{\sigma}_{w_{j'} g_{j'}}(\mathsf{t})} = \frac{\widehat{\sigma}_{g_j}(\mathsf{t})}{2^{d_0} \sum_{j'} \widehat{\sigma}_{g_{j'}}(\mathsf{t})}.$$

We are now in a position to prove (A.2). Since a maximum is greater than an average (with respect to the coordinate index $j$ and sign vector $\mathbf{w}$), we have

$$\max_{j=1,2,\ldots,d} \widehat{\rho}^2(g_j(X_j), Y \mid \mathbf{X} \in \mathsf{t}) = \max_{j=1,2,\ldots,d} \widehat{\rho}^2(w_j g_j(X_j), Y \mid \mathbf{X} \in \mathsf{t})$$

$$\geq \sum_{(j, \mathbf{w})} \pi(j, \mathbf{w}) \widehat{\rho}^2(w_j g_j(X_j), Y \mid \mathbf{X} \in \mathsf{t}).$$

Jensen's inequality for the square function yields

$$\sum_{(j, \mathbf{w})} \pi(j, \mathbf{w}) \widehat{\rho}^2(w_j g_j(X_j), Y \mid \mathbf{X} \in \mathsf{t}) \geq \sum_{\mathbf{w}} \pi(\mathbf{w}) |\sum_j \pi(j \mid \mathbf{w}) \widehat{\rho}(w_j g_j(X_j), Y \mid \mathbf{X} \in \mathsf{t})|^2$$

$$= \sum_{\mathbf{w}} \pi(\mathbf{w}) \frac{\widehat{\sigma}_{Y_0}^2(\mathsf{t})}{(\sum_{j'} \widehat{\sigma}_{g_{j'}}(\mathsf{t}))^2} \widehat{\rho}^2(Y_0, Y \mid \mathbf{X} \in \mathsf{t})$$

$$\geq \frac{\sum_{\mathbf{w}} \pi(\mathbf{w}) \widehat{\sigma}_{Y_0}^2(\mathsf{t})}{(\sum_{j'} \widehat{\sigma}_{g_{j'}}(\mathsf{t}))^2} \min_{\mathbf{w}} \widehat{\rho}^2(Y_0, Y \mid \mathbf{X} \in \mathsf{t}) \qquad \text{(A.4)}$$

Next, note that $\sum_{\mathbf{w}} \pi(\mathbf{w}) \widehat{\sigma}_{Y_0}^2(\mathsf{t}) = \sum_j \widehat{\sigma}_{g_j}^2(\mathsf{t})$, since the covariance terms of $\widehat{\sigma}_{Y_0}^2(\mathsf{t})$ have mean zero with respect to $\pi(\mathbf{w}) \equiv 2^{-d_0}$; that is,

$$\sum_{\mathbf{w}} \pi(\mathbf{w}) \widehat{\sigma}_{Y_0}^2(\mathsf{t})$$

$$= \sum_{\mathbf{w}} \sum_j \pi(\mathbf{w}) \widehat{\sigma}_{w_j g_j}^2(\mathsf{t}) + \sum_{\mathbf{w}} \sum_j \pi(\mathbf{w}) \widehat{\mathrm{COV}}(w_j g_j(X_j), w_{j'} g_{j'}(X_{j'}) \mid \mathbf{X} \in \mathsf{t})$$

$$= \sum_j \widehat{\sigma}_{g_j}^2(\mathsf{t}) \sum_{\mathbf{w}} \pi(\mathbf{w}) + \sum_{j,j'} \widehat{\mathrm{COV}}(g_j(X_j), g_{j'}(X_{j'}) \mid \mathbf{X} \in \mathsf{t}) \sum_{\mathbf{w}} \pi(\mathbf{w}) w_j w_{j'}$$

$$= \sum_j \widehat{\sigma}_{g_j}^2(\mathsf{t}).$$

Combining this with (A.4) shows that $\max_{j=1,2,\ldots,d} \widehat{\rho}^2(g_j(X_j), Y \mid \mathbf{X} \in \mathsf{t})$ is at least

$$\frac{\sum_j \widehat{\sigma}_{g_j}^2(\mathsf{t})}{(\sum_{j'} \widehat{\sigma}_{g_{j'}}(\mathsf{t}))^2} \min_{\mathbf{w}} \widehat{\rho}^2(Y_0, Y \mid \mathbf{X} \in \mathsf{t}) \geq \frac{\min_{\mathbf{w}} \widehat{\rho}^2(Y_0, Y \mid \mathbf{X} \in \mathsf{t})}{d_0},$$

where the last inequality follows from the Cauchy-Schwarz inequality. If each $g_j(\cdot)$ has nonnegative Pearson correlation with the others in the node, then $\widehat{\sigma}_{Y_0}^2(\mathsf{t}) \geq \sum_j \widehat{\sigma}_{g_j}^2(\mathsf{t})$ and thus the same argument as above can be repeated with $Y_0 = \sum_j g_j(X_j)$ to prove (A.3). $\qquad \square$

*Proof of Theorem 1.* Let $\overline{\mathrm{Err}}(\widehat{Y}) = \frac{1}{n} \sum_{i=1}^n (Y_i' - \widehat{Y}(T, \mathbf{X}_i'))^2$ denote the test error of $\widehat{Y}(T)$ on a test sample $\mathcal{D}_n' = \{(\mathbf{X}_i', Y_i')\}_{i=1}^n$ of size $n$. Let $\mathcal{T}_{\mathbf{X}, \mathbf{X}'}$ denote the collection of tree-structured partitions constructed on the grid $\{\mathbf{X}_i\}_{i=1}^n \cup \{\mathbf{X}_i'\}_{i=1}^n$ with $2n$ points. Note that the VC-dimension of the collection of axis-parallel splits is at most the VC-dimension of the collection of all half-spaces, namely, $d+1$. In this case, Lemma B.2 in [1] shows that the number of trees in $\mathcal{T}_{\mathbf{X}, \mathbf{X}'}$ with exactly $|T|$ nodes is at most $(2ne/(d+1))^{|T|(d+1)}$. Using this, we have

$$\sum_{T \in \mathcal{T}_{\mathbf{X}, \mathbf{X}'}} e^{-L(T)} \leq \sum_{k:|T|=k\geq 1} \exp\Big( -L(T) + |T|(d+1) \log(2ne/(d+1)) \Big) \leq 1,$$

if $L(T)$ is any penalty that exceeds $2|T|(d+1) \log(2en/(d+1)) \geq |T|(\log(2) + (d+1) \log(2ne/(d+1)))$. Thus, a penalty equal to $L(T) := 2|T|(d+1) \log(2en/(d+1)) \geq |T|(\log(2) + (d+$

1) $\log(2ne/(d+1)))$ satisfies Kraft's inequality, i.e., $\sum_{T \in \mathcal{T}_{\mathbf{X},\mathbf{X}'}} e^{-L(T)} \leq 1$. Observe also that $\mathcal{T}_{\mathbf{X},\mathbf{X}'}$ is symmetric in the pairs $(\mathbf{X}_i, \mathbf{X}'_i)$. By Lemma 2.1 in [3], for all $\gamma > 0$,

$$\mathbb{P}\left( \max_{T \in \mathcal{T}_{\mathbf{X},\mathbf{X}'}} \frac{\overline{\mathrm{Err}}(\widehat{Y}(T)) - \overline{\mathrm{err}}(\widehat{Y}(T))}{\frac{1}{n\gamma^2}(L(T) + \log(2/\delta)) + \frac{1}{2}S^2(\widehat{Y}(T))} < \gamma \right) \geq 1 - \delta/2, \qquad (A.5)$$

where $S^2(\widehat{Y}(T)) = \frac{1}{n}\sum_{i=1}^{n}((Y'_i - \widehat{Y}(\mathbf{X}'_i))^2 - (Y_i - \widehat{Y}(\mathbf{X}_i))^2)^2$. Using the fact that $S^2(\widehat{Y}(T)) \leq 8B^2(\overline{\mathrm{Err}}(\widehat{Y}(T)) + \overline{\mathrm{err}}(\widehat{Y}(T)))$ and $\widehat{T} \in \mathcal{T}_{\mathbf{X},\mathbf{X}'}$, and choosing $\gamma^{-1} = 12B^2$, we find that

$$\overline{\mathrm{Err}}(\widehat{Y}(\widehat{T})) \leq 2\overline{\mathrm{err}}(\widehat{Y}(\widehat{T})) + \frac{18B^2 L(\widehat{T})}{n} + \frac{18B^2 \log(2/\delta)}{n} \qquad (A.6)$$

occurs with probability at least $1 - \delta/2$. Next, using Lemma 9 from [2] together with the bound $\overline{\mathrm{Err}}(\widehat{Y}(T)) \leq 4B^2$ and the Kraft summability of the penalty $L(T)$, we have that for all $\gamma > 0$,

$$\mathbb{P}\left( \max_{T \in \mathcal{T}_{\mathbf{X}}} \frac{\mathrm{Err}(\widehat{Y}(T)) - \overline{\mathrm{Err}}(\widehat{Y}(T))}{\frac{4B^2}{n\gamma^2}(L(T) + \log(2/\delta)) + \mathrm{Err}(\widehat{Y}(T)) + \overline{\mathrm{Err}}(\widehat{Y}(T))} < \gamma \right) \geq 1 - \delta/2,$$

where $\mathcal{T}_{\mathbf{X}} \subset \mathcal{T}_{\mathbf{X},\mathbf{X}'}$ is the set of all tree-structured partitions constructed using the grid $\{\mathbf{X}_i\}_{i=1}^{n}$. Choose $\gamma = 1/3$. Since $\widehat{T} \in \mathcal{T}_{\mathbf{X}}$, with probability at least $1 - \delta/2$,

$$\mathrm{Err}(\widehat{Y}(\widehat{T})) \leq 2\overline{\mathrm{Err}}(\widehat{Y}(\widehat{T})) + \frac{18B^2 L(\widehat{T})}{n} + \frac{18B^2 \log(2/\delta)}{n}. \qquad (A.7)$$

Combining (A.6) and (A.7), we have that with probability at least $1 - \delta$,

$$\mathrm{Err}(\widehat{Y}(\widehat{T})) \leq 4R_\alpha(\widehat{Y}(\widehat{T})) + \frac{54B^2 \log(2/\delta)}{n},$$

provided $d > (n+1)/2$ and $\alpha > \frac{27B^2(d+1)\log(2en/(d+1))}{n}$. The conclusion of the theorem follows from the definition of $\widehat{T}$ as a minimizer of $R_\alpha(\widehat{Y}(T))$. $\qquad \square$

*Proof of Theorem 2.* The identity (10) is shown by first noting that, in the special case of uniform $\mathbf{X}$, the probability $\mathbb{P}(X \leq s^* \mid \mathbf{X} \in \mathsf{t})$ from Lemma B.1 in Appendix B is equal to $(s^* - a)/(b - a)$. Rearranging the resulting expression yields the desired identity. $\qquad \square$

*Proof of Lemma 1.* We first prove (11) for a general decision stump $\widetilde{Y}$. The training error in $\mathsf{t}$ after splitting is

$$\frac{1}{N(\mathsf{t})}\sum_{\mathbf{X}_i \in \mathsf{t}}(Y_i - \widetilde{Y}_i)^2 = \frac{1}{N(\mathsf{t})}\sum_{\mathbf{X}_i \in \mathsf{t}_L}(Y_i - \overline{Y}_{\mathsf{t}_L})^2 + \frac{1}{N(\mathsf{t})}\sum_{\mathbf{X}_i \in \mathsf{t}_R}(Y_i - \overline{Y}_{\mathsf{t}_R})^2$$

$$= \widehat{\Delta}(\mathsf{t})\left(1 - \frac{\widehat{\Delta}(s,\mathsf{t})}{\widehat{\Delta}(\mathsf{t})}\right)$$

$$= \frac{1}{N(\mathsf{t})}\sum_{\mathbf{X}_i \in \mathsf{t}}(Y_i - \overline{Y}_{\mathsf{t}})^2 \times (1 - \widehat{\rho}^2(\widetilde{Y}, Y \mid \mathbf{X} \in \mathsf{t})),$$

where the last equality follows from Lemma A.1. Finally, $1 - \widehat{\rho}^2(\widetilde{Y}, Y \mid \mathbf{X} \in \mathsf{t}) \leq \exp(-\widehat{\rho}^2(\widetilde{Y}, Y \mid \mathbf{X} \in \mathsf{t}))$ follows from $1 - z \leq e^{-z}$ for $z \geq 0$. To show (12), we use (11) with $\widetilde{Y} = \widehat{Y}$ recursively together with the identity

$$\overline{\mathrm{err}}(\widehat{Y}(T_K)) = \sum_{\mathsf{t}} \widehat{P}(\mathsf{t})\widehat{\Delta}(\mathsf{t}),$$

where the sum extends over all terminal nodes $\mathsf{t}$ of $T_K$. We stop once we reach the root node, at which point the training error is simply $\widehat{\sigma}_Y^2$. $\qquad \square$

*Proof of Fact 1.* Fact 1 is a special case of the following lemma. In order to state the lemma, we will need to introduce the concept of stationary intervals. We define a *stationary interval* of a univariate function $g(\cdot)$ to be a maximal interval $I$ such that $g(I) = c$, where $c$ is a local extremum of $g(\cdot)$ ($I$ is maximal in the sense that there does not exist an interval $I'$ such that $I \subset I'$ and $g(I') = c$). In particular, note that a monotone function does not have any stationary intervals.

**Lemma A.4.** *Almost surely, uniformly over all step functions $g(\cdot)$ of $X$ that have at most $V$ constant pieces and $M$ stationary intervals in the node, we have*

$$\widehat{\rho}(\widehat{Y}, Y \mid \mathbf{X} \in \mathrm{t}) \geq \frac{1}{\sqrt{D^{-1}MN(\mathrm{t}) + (V - M - 1) \wedge (1 + \log(2N(\mathrm{t})))}} \times |\widehat{\rho}(g(X), Y \mid \mathbf{X} \in \mathrm{t})|,$$

(A.8)

*where $D \geq 1$ is the smallest number of data points in a stationary interval of $g(\cdot)$ that contains at least one data point.*[2]

*Proof of Lemma A.4.* Let $g(\cdot)$ be any function of a generic coordinate $X$ and assume that the data points in the node are labeled for simplicity as $\{X_i : \mathbf{X} \in \mathrm{t}\} = \{X_1, X_2, \ldots, X_{N(\mathrm{t})}\}$ and ordered such that $X_1 \leq X_2 \leq \cdots \leq X_{N(\mathrm{t})}$. Without loss of generality, $g(\cdot)$ can be redefined to linearly interpolate between the values $g(X_1), g(X_2), \ldots, g(X_{N(\mathrm{t})})$. We look at the (empirical Bayesian) prior $\Pi$ on splits $s$ with density

$$\frac{d\Pi(s)}{ds} = \frac{|g'(s)|\sqrt{\widehat{P}(\mathrm{t}_L)\widehat{P}(\mathrm{t}_R)}}{\int |g'(s')|\sqrt{\widehat{P}(\mathrm{t}_L)\widehat{P}(\mathrm{t}_R)}ds'},$$

where we remind the reader that $\widehat{P}(\mathrm{t}_L) = 1 - \widehat{P}(\mathrm{t}_R) = \frac{1}{N(\mathrm{t})} \sum_{\mathbf{X}_i \in \mathrm{t}} \mathbf{1}(X_i \leq s)$. Here, $g'(s)$ equals the divided difference $\frac{g(X_{i+1}) - g(X_i)}{X_{i+1} - X_i}$ when $X_i \leq s < X_{i+1}$, $i = 1, 2, \ldots N(\mathrm{t}) - 1$. Accordingly, observe that $\Pi$ has a piecewise constant density with knots at the data points and supported between the minimum and maximum of the data $X_i$. Since, by definition, $\widehat{Y}$ maximizes $s \mapsto \widehat{\rho}(\widetilde{Y}, Y \mid \mathbf{X} \in \mathrm{t})$ and a maximum is larger than an average, we have

$$\widehat{\rho}(\widehat{Y}, Y \mid \mathbf{X} \in \mathrm{t}) = \max_s \widehat{\rho}(\widetilde{Y}, Y \mid \mathbf{X} \in \mathrm{t})$$

$$\geq \int \widehat{\rho}(\widetilde{Y}, Y \mid \mathbf{X} \in \mathrm{t})d\Pi(s) = \int \sqrt{\frac{\widehat{\Delta}(s, \mathrm{t})}{\Delta(\mathrm{t})}}d\Pi(s),$$

(A.9)

where the last equality follows from Lemma A.1. Next, working from the representation (4), note that the reduction in impurity admits the form

$$\widehat{\Delta}(s, \mathrm{t}) = \left(\frac{1}{\sqrt{\widehat{P}(\mathrm{t}_L)\widehat{P}(\mathrm{t}_R)}}\left(\frac{1}{N(\mathrm{t})}\sum_{\mathbf{X}_i \in \mathrm{t}}(\mathbf{1}(s < X_i) - \widehat{P}(\mathrm{t}_R))(Y_i - \overline{Y}_\mathrm{t})\right)\right)^2,$$

(A.10)

and, hence, integrating inside the square in (A.10) against $g'(s)\sqrt{\widehat{P}(\mathrm{t}_L)\widehat{P}(\mathrm{t}_R)}$, we have

$$\int g'(s)\left(\frac{1}{N(\mathrm{t})}\sum_{\mathbf{X}_i \in \mathrm{t}}(\mathbf{1}(s < X_i) - \widehat{P}(\mathrm{t}_R))(Y_i - \overline{Y}_\mathrm{t})\right)ds$$

$$= \frac{1}{N(\mathrm{t})}\sum_{\mathbf{X}_i \in \mathrm{t}}(g(X_i) - \frac{1}{N(\mathrm{t})}\sum_{\mathbf{X}_{i'} \in \mathrm{t}}g(X_{i'}))(Y_i - \overline{Y}_\mathrm{t})$$

$$= \widehat{\mathrm{COV}}(g(X), Y \mid \mathbf{X} \in \mathrm{t}).$$

(A.11)

Using the inequality (A.9) together with the identities (A.10) and (A.11), we have

$$\widehat{\rho}(\widehat{Y}, Y \mid \mathbf{X} \in \mathrm{t}) \geq \int \sqrt{\frac{\widehat{\Delta}(s, \mathrm{t})}{\Delta(\mathrm{t})}}d\Pi(s)$$

$$\geq \frac{\sqrt{\widehat{\mathrm{VAR}}(g(X) \mid \mathbf{X} \in \mathrm{t})}}{\int |g'(s)|\sqrt{\widehat{P}(\mathrm{t}_L)\widehat{P}(\mathrm{t}_R)}ds} \times |\widehat{\rho}(g(X), Y \mid \mathbf{X} \in \mathrm{t})|.$$

(A.12)

Therefore, from (A.12), we are led to determine how small the ratio

$$\frac{\sqrt{\widehat{\text{VAR}}(g(X) \mid \mathbf{X} \in \mathsf{t})}}{\int |g'(s)| \sqrt{\widehat{P}(\mathsf{t}_L)\widehat{P}(\mathsf{t}_R)} ds}. \tag{A.13}$$

can be, ideally in terms of some simple structural characteristics of $g(\cdot)$. Our next task is to simplify (A.13) so that its numerator and denominator can be more easily compared. To this end, observe that

$$
\begin{aligned}
&\int |g'(s)| \sqrt{\widehat{P}(\mathsf{t}_L)\widehat{P}(\mathsf{t}_R)} ds \\
&= \sum_{i=0}^{N(\mathsf{t})} \int_{N(\mathsf{t})\widehat{P}(\mathsf{t}_L)=i} |g'(s)| \sqrt{\frac{i}{N(\mathsf{t})}\left(1 - \frac{i}{N(\mathsf{t})}\right)} ds \\
&= \sum_{i=1}^{N(\mathsf{t})-1} \int_{X_i}^{X_{i+1}} |g'(s)| ds \sqrt{\frac{i}{N(\mathsf{t})}\left(1 - \frac{i}{N(\mathsf{t})}\right)} \\
&= \frac{1}{N(\mathsf{t})} \sum_{i=1}^{N(\mathsf{t})-1} |g(X_{i+1}) - g(X_i)| \sqrt{i(N(\mathsf{t}) - i)}, \tag{A.14}
\end{aligned}
$$

where the penultimate equality follows from the fact that $\widehat{P}(\mathsf{t}_L) = i/N(\mathsf{t})$ if and only if $X_i \le s < X_{i+1}$. Next, we further simplify the above expression (A.14) using summation by parts, that is,

$$\frac{1}{N(\mathsf{t})} \sum_{i=1}^{N(\mathsf{t})-1} |g(X_{i+1}) - g(X_i)| \sqrt{i(N(\mathsf{t}) - i)} = -\frac{1}{N(\mathsf{t})} \sum_{i=1}^{N(\mathsf{t})} g(X_i)(b_i - b_{i-1}), \tag{A.15}$$

where $b_i = \text{sgn}(g(X_{i+1}) - g(X_i)) \times \sqrt{i(N(\mathsf{t}) - i)}$ with $b_0 = b_{N(\mathsf{t})} = 0$. Next, since $\sum_{i=1}^{N(\mathsf{t})}(b_i - b_{i-1}) = b_{N(\mathsf{t})} - b_0 = 0$, (A.15) can be written as

$$-\frac{1}{N(\mathsf{t})} \sum_{i=1}^{N(\mathsf{t})} \left(g(X_i) - \frac{1}{N(\mathsf{t})} \sum_{\mathbf{X}_{i'} \in \mathsf{t}} g(X_{i'})\right)(b_i - b_{i-1}). \tag{A.16}$$

Moreover, we can express the variance $\widehat{\text{VAR}}(g(X) \mid \mathbf{X} \in \mathsf{t})$ in a similar form, viz.,

$$\widehat{\text{VAR}}(g(X) \mid \mathbf{X} \in \mathsf{t}) = \frac{1}{N(\mathsf{t})} \sum_{i=1}^{N(\mathsf{t})} \left(g(X_i) - \frac{1}{N(\mathsf{t})} \sum_{\mathbf{X}_{i'} \in \mathsf{t}} g(X_{i'})\right)^2. \tag{A.17}$$

To obtain the best lower bound on the ratio (A.13), we attempt to solve the program

$$\min_{g(\cdot) \in \mathcal{G}} \frac{\widehat{\text{VAR}}(g(X) \mid \mathbf{X} \in \mathsf{t})}{\left(\int |g'(s)| \sqrt{\widehat{P}(\mathsf{t}_L)\widehat{P}(\mathsf{t}_R)} ds\right)^2}, \tag{A.18}$$

where $\mathcal{G}$ is a collection of functions. In light of the expressions (A.16) and (A.17), the program (A.18) is equivalent to the following program:

$$\min_{\mathbf{a} \in \mathcal{A}} \sum_{i=1}^{N(\mathsf{t})} |a_i|^2 \quad \text{s.t.} \quad \frac{1}{\sqrt{N(\mathsf{t})}} \sum_{i=1}^{N(\mathsf{t})} a_i(b_i - b_{i-1}) = 1, \quad \sum_{i=1}^{N(\mathsf{t})} a_i = 0. \tag{A.19}$$

where $b_i = \text{sgn}(a_{i+1} - a_i) \sqrt{i(N(\mathsf{t}) - i)}$ and $\mathcal{A}$ is a collection of vectors in $\mathbb{R}^{N(\mathsf{t})}$. In order to incorporate structural and/or regularity properties of $g(\cdot)$, we will need to impose conditions on $\mathcal{G}$ or, since we associate $a_i$ with $g(X_i) - \frac{1}{N(\mathsf{t})} \sum_{\mathbf{X}_{i'} \in \mathsf{t}} g(X_{i'})$, on $\mathcal{A}$. However, not all specifications make the program tractable to solve, or even convex. As a compromise, we fix the signs of the $b_i$ in advance. That is, we specify three additional constraints, namely, $b_i = 0$, $b_i > 0$, and $b_i < 0$— corresponding to locations where $g(\cdot)$ is constant, increasing, and decreasing, respectively—and solve the resulting (quadratic) program. More formally, let $V$ and $M$ respectively denote the number

of constant pieces and stationary intervals of $g(\cdot)$ and let $S = \{i_k\}_{1 \le k \le V-1}$ and $S' \subset S$ be two subsets of $\{1, 2, \ldots, N(\mathrm{t}) - 1\}$ with $i_0 = 0$ and $i_V = N(\mathrm{t})$. Let $\mathcal{A} = \{\mathbf{a} \in \mathbb{R}^{N(\mathrm{t})} : b_i = 0 \text{ for } i \notin S, \ b_i > 0 \text{ for } i \in S', \ b_i < 0 \text{ for } i \notin S'\}$, and $D_k = i_k - i_{k-1}$. (Note that $M$ can be regarded as the number of times $g(\cdot)$ changes from strictly increasing to decreasing (or vice versa) and hence $b_{i-1}b_i < 0$ at most $M$ times.) With these specifications fixed, the program (A.19) becomes

$$\min_{\mathbf{a} \in \mathcal{A}} \quad \sum_{k=1}^{V} |a_{i_k}|^2 D_k \quad \text{s.t.} \quad \frac{1}{\sqrt{N(\mathrm{t})}} \sum_{k=1}^{V} a_{i_k}(b_{i_k} - b_{i_{k-1}}) = 1, \quad \sum_{k=1}^{V} a_{i_k} D_k = 0. \tag{A.20}$$

Using the method of Lagrange multipliers, it is easy to see that the solution to (A.20) is

$$a_{i_k}^* = \frac{\sqrt{N(\mathrm{t})}(b_{i_k} - b_{i_{k-1}})/D_k}{\sum_{k=1}^{V}(b_{i_k} - b_{i_{k-1}})^2/D_k}, \quad k = 1, 2, \ldots, V, \tag{A.21}$$

and the value of the program is

$$\frac{N(\mathrm{t})}{\sum_{k=1}^{V}(b_{i_k} - b_{i_{k-1}})^2/D_k}. \tag{A.22}$$

Lemma B.3 in Appendix B shows that (A.22) is at least

$$\frac{1}{D^{-1}MN(\mathrm{t}) + (V - M - 1) \wedge (1 + \log(2N(\mathrm{t})))},$$

where $D$ is the smallest number of data points in a stationary interval of $g(\cdot)$ that contains at least one data point. Hence by (A.12), we obtain the desired (A.8). $\qquad\square$

Fact 1 follows immediately from (A.8) by noting that, in this case, $M = 0$. $\qquad\square$

**Remark A.1.** *Another candidate prior $\Pi$ for (A.9) is*

$$\frac{d\Pi(j,s)}{d(j,s)} := \frac{|g_j'(s)|\sqrt{\widehat{P}_j(\mathrm{t}_L)\widehat{P}_j(\mathrm{t}_R)}}{\sum_j \int |g_j'(s')|\sqrt{\widehat{P}_j(\mathrm{t}_L)\widehat{P}_j(\mathrm{t}_R)}ds'},$$

*which, akin to (A.12), leads to the correlation inequality*

$$\widehat{\rho}(\widehat{Y}, Y \mid \mathbf{X} \in \mathrm{t}) \ge \frac{\sqrt{\widehat{VAR}(\sum_j g_j(X_j) \mid \mathbf{X} \in \mathrm{t})}}{\sum_j \int |g_j'(s)|\sqrt{\widehat{P}_j(\mathrm{t}_L)\widehat{P}_j(\mathrm{t}_R)}ds} \times |\widehat{\rho}(\sum_j g_j(X_j), Y \mid \mathbf{X} \in \mathrm{t})|.$$

*While this enables comparisons with additive models via $\widehat{\rho}(\sum_j g_j(X_j), Y \mid \mathbf{X} \in \mathrm{t})$, the factor $\frac{\sqrt{\widehat{VAR}(\sum_j g_j(X_j)|\mathbf{X}\in\mathrm{t})}}{\sum_j \int |g_j'(s)|\sqrt{\widehat{P}_j(\mathrm{t}_L)\widehat{P}_j(\mathrm{t}_R)}ds}$ is less amenable to analysis.*

*Proof of Theorem 3.* We first employ a technique similar to (A.12) in the proof of Fact 1 (essentially, the infinite sample analog) to lower bound $\rho^2(\widehat{Y}^*, Y \mid \mathbf{X} \in \mathrm{t})$. That is, for each function $g(\cdot)$ of $X$ and node t,

$$\rho^2(\widehat{Y}^*, Y \mid \mathbf{X} \in \mathrm{t}) \ge \Lambda \times \rho^2(g(X), Y \mid \mathbf{X} \in \mathrm{t}), \tag{A.23}$$

where

$$\Lambda := \frac{\mathrm{VAR}(g(X) \mid X \in [a,b])}{\left(\int_a^b |g'(s)|\sqrt{\frac{s-a}{b-a}\frac{b-s}{b-a}}ds\right)^2}.$$

In contrast with the proof of Fact 1, here we do not attempt to minimize $\Lambda$ over all $g(\cdot)$ in some function class. Rather, we attempt to lower bound it for a *fixed* $g(\cdot)$. Now, (A.23) is valid for all $g_j(X_j)$ and so we can instead consider the maximum correlation over all $g_j(X_j)$, i.e., $\max_j \rho^2(g_j(X_j), Y \mid \mathbf{X} \in \mathrm{t})$, where now $\Lambda$ is the minimum over all $g_j(X_j)$. By the infinite sample analog of (A.3) in Lemma A.3, we have $\max_j \rho^2(g_j(X_j), Y \mid \mathbf{X} \in \mathrm{t}) \ge \frac{\rho^2(Y, Y|\mathbf{X}\in\mathrm{t})}{d_0} = 1/d_0$, and hence

$$\rho^2(\widehat{Y}^*, Y \mid \mathbf{X} \in \mathrm{t}) \ge \Lambda/d_0. \tag{A.24}$$

Next, we show that $\Lambda$ can be further lower bounded by a positive constant that is independent of t. To this end, note that $\Lambda$ is continuous in $(a, b)$ and strictly positive for all $a < b$ and, furthermore by Lemma B.2 in Appendix B,

$$\inf_c \liminf_{(a,b) \to (c,c)} \Lambda = \Omega(1/R),$$

where $R = \sup_{c \in [0,1]} \inf\{r \geq 1 : g^{(r)}(\cdot)$ exists and is continuous and nonzero at $c\}$—which means that $\inf_{(a,b)} \Lambda > 0$. Note that, in particular, $R$ is finite if $g(\cdot)$ admits a power series representation. Taking the minimum of $\inf_{(a,b)} \Lambda$ over all $g_j(\cdot)$—each of which has finite $R$—results in a positive quantity that depends only on each $g_j(\cdot)$ individually. This shows that $\inf_t \rho^2(\widehat{Y}^*, Y \mid \mathbf{X} \in t) \geq C/d_0$ for some positive constant $C$ that depends only on each $g_j(\cdot)$ individually and not on $d_0$. Next, we will show that, almost surely,

$$\liminf_n \widehat{\rho}_{\mathcal{H}}^2 = \liminf_n \inf_t \widehat{\rho}^2(\widehat{Y}, Y \mid \mathbf{X} \in t) \geq \inf_t \rho^2(\widehat{Y}^*, Y \mid \mathbf{X} \in t),$$

from which the first statement in Theorem 3 will follow, i.e., $\liminf_n \widehat{\rho}_{\mathcal{H}}^2 \geq C/d_0$ almost surely. First, by definition of $\widehat{Y}$ as the optimizer of $(j, s) \mapsto \widehat{\rho}^2(\widetilde{Y}, Y \mid \mathbf{X} \in t)$, almost surely,

$$\liminf_n \inf_t \widehat{\rho}^2(\widehat{Y}, Y \mid \mathbf{X} \in t) \geq \liminf_n \inf_t \widehat{\rho}^2(\widehat{Y}^*, Y \mid \mathbf{X} \in t),$$

where we remind the reader that $\widehat{Y}^*$ is the decision stump $\widetilde{Y}$ at an optimal theoretical direction $j^*$ and split $s^*$. Next, note that $\widehat{\rho}(\widehat{Y}^*, Y \mid \mathbf{X} \in t)$ is invariant to scale. Working instead with $\frac{N(t)}{n} \widehat{Y}^*$ and $\frac{N(t)}{n} Y$, we find that the correlation involves terms (empirical processes) of the form $\frac{1}{n} \sum_{i=1}^n \mathbf{1}(\mathbf{X}_i \in t')$, $\frac{1}{n} \sum_{i=1}^n \mathbf{1}(\mathbf{X}_i \in t') Y_i$, and $\frac{1}{n} \sum_{i=1}^n \mathbf{1}(\mathbf{X}_i \in t) Y_i^2$, where $t'$ is either the parent node $t$ or one of the daughter nodes, $t_L^* := \{\mathbf{X} \in t : X \leq s^*\}$ and $t_R^* := \{\mathbf{X} \in t : X > s^*\}$ at an optimal theoretical split $s^*$. The collection of hyperrectangles in $\mathbb{R}^d$ is a finite VC-class with VC-dimension at most $2d$, and hence these terms converge almost surely, uniformly over all nodes $t'$, to their respective population level counterparts when $d = o(n)$. Thus, $\liminf_n \inf_t \widehat{\rho}^2(\widehat{Y}^*, Y \mid \mathbf{X} \in t) \overset{\text{a.s.}}{=} \inf_t \liminf_n \widehat{\rho}^2(\widehat{Y}^*, Y \mid \mathbf{X} \in t) \overset{\text{a.s.}}{=} \inf_t \rho^2(\widehat{Y}^*, Y \mid \mathbf{X} \in t)$.

The almost sure limit (19) in Theorem 3 follows from (17) with $\delta = 1/n^2$ and $\liminf_n \widehat{\rho}_{\mathcal{H}}^2 \geq C/d_0$ (almost surely) together with the Borel-Cantelli lemma. $\square$

*Proof of Theorem 4.* As mentioned right before the statement of Theorem 4, we need to prove (20). To lighten notation, we consider a generic direction $X$, write $N$ for $N(t)$, and assume that the data is labeled in the node $t$ so that $X_1 \leq X_2 \leq \cdots \leq X_N$. Let $I$ be one of the intervals on which $g(X)$ is constant and let $X_{i_1} = \min\{X_i \in I : \mathbf{X}_i \in t\}$ and $X_{i_2} = \max\{X_i \in I : \mathbf{X}_i \in t\}$ so that $i_1 \leq i_2$. We will show that if $\widehat{\Delta}(\hat{s}, t) > 0$, then the maximum of $\widehat{\Delta}(s, t)$ for $s \in [X_{i_1}, X_{i_2+1})$ must occur at the boundary, i.e., $[X_{i_1}, X_{i_1+1})$ or $[X_{i_2}, X_{i_2+1})$. Let $\mu_1 = \frac{1}{i_1} \sum_{\mathbf{X} \in t, X_i \leq X_{i_1}} Y_i$, $\mu_2 = \frac{1}{N-i_2} \sum_{\mathbf{X} \in t, X_i > X_{i_2}} Y_i$, and $\mu = \frac{1}{i_2 - i_1} \sum_{X_{i_1} < X_i \leq X_{i_2}} Y_i$. Suppose $X_i \leq s < X_{i+1}$. Then the decrease in impurity equals

$$\widehat{\Delta}(s, t) = \frac{i}{N} \times \frac{N-i}{N} \times \left( \frac{1}{i}(i_1 \mu_1 + (i - i_1)\mu) - \frac{1}{N-i}((N - i_2)\mu_2 + (i_2 - i)\mu) \right)^2.$$

Viewed as a function of $i$, $\widehat{\Delta}(s, t) = \widehat{\Delta}(i)$ has two critical values, one of which is a zero solution, namely, $i^* = \frac{(\mu_1 - \mu)i_1 N}{\mu_1 i_1 + \mu_2(N - i_2) - \mu(N + i_1 - i_2)}$. The other critical value, equal to

$$i^* = \frac{(\mu_1 - \mu)i_1 N}{\mu_1 i_1 - \mu_2(N - i_2) + \mu(N - i_1 - i_2)},$$

produces the value

$$\widehat{\Delta}(i^*) = \frac{4 i_1 (N - i_2)(\mu_1 - \mu)(\mu - \mu_2)}{N^2}.$$

We will be done if we can show that either

$$\widehat{\Delta}(i_1) = \frac{i_1(\mu_1(N - i_1) - \mu_2(N - i_2) - \mu(i_2 - i_2))^2}{N^2(N - i_1)}$$

or

$$\widehat{\Delta}(i_2) = \frac{(N - i_2)(\mu_1 i_1 - \mu_2 i_2 + \mu(i_2 - i_1))^2}{N^2 i_2}$$

are (strictly) greater than $\widehat{\Delta}(i^*)$. After some tedious algebra, we find that $\widehat{\Delta}(i_1) > \widehat{\Delta}(i^*)$ and $\widehat{\Delta}(i_2) > \widehat{\Delta}(i^*)$ with equality if and only if $i^* = i_1$ and $i^* = i_2$, respectively. □

*Proof of Theorem 5.* We first show that

$$\overline{\mathrm{err}}(\widehat{Y}(T_K)) \leq \widehat{\sigma}_Y^2 \exp\Big( - \widehat{\rho}_{\mathcal{M}}^2 \sum_{k=1}^{K} (\log_2(4N_k))^{-1} \Big). \tag{A.25}$$

By (11) in Lemma 1, the training error in the node is decreased by a factor of $\exp(-\widehat{\rho}^2(\widehat{Y}, Y \mid \mathbf{X} \in$ t)) each time the node is split. By Fact 1, almost surely, $\widehat{\rho}^2(\widehat{Y}, Y \mid \mathbf{X} \in t) \geq \frac{1}{1 + \log(2N(t))} \times \widehat{\rho}_{\mathcal{M}}^2 \geq \frac{1}{\log_2(4N(t))} \times \widehat{\rho}_{\mathcal{M}}^2 \geq \frac{1}{\log_2(4N_k)} \times \widehat{\rho}_{\mathcal{M}}^2$, if t is a node at level $k$. Thus, the training error at level $k+1$ is at most $\exp(-\widehat{\rho}_{\mathcal{M}}^2 (\log_2(4N_k))^{-1})$ times the training error at level $k$—in other words, the training error is geometrically decreasing. The proof of (A.25) can then be completed using an induction argument, noting that the training error at the root node is simply $\widehat{\sigma}_Y^2$.

For the training error bound (22), we use the inequality $\sum_{k=1}^{K} \frac{1}{\log_2(4Ank^a/2^k)} \geq \log\Big( \frac{\log_2(4K^a An)}{\log_2(4K^a An) - K} \Big)$ for integers $K \geq 1$. By (A.25), if $T_K$ is a fully grown tree of depth $K$, then under Assumption 1, i.e., $N_k \leq Ank^a/2^k$, we have

$$\overline{\mathrm{err}}(\widehat{Y}(T_K)) \leq \widehat{\sigma}_Y^2 \exp\Big( - \widehat{\rho}_{\mathcal{M}}^2 \sum_{k=1}^{K} (\log_2(4N_k))^{-1} \Big)$$

$$\leq \widehat{\sigma}_Y^2 \exp\Big( - \widehat{\rho}_{\mathcal{M}}^2 \sum_{k=1}^{K} \frac{1}{\log_2(4Ank^a/2^k)} \Big)$$

$$\leq \widehat{\sigma}_Y^2 \Big( 1 - \frac{K}{\log_2(4K^a An)} \Big)^{\widehat{\rho}_{\mathcal{M}}^2}. \tag{A.26}$$

Next, we show (23), i.e., the bound on the prediction error. By Theorem 1, with high probability, the leading behavior of the test error $\mathrm{Err}(\widehat{Y}(\widehat{T}))$ is governed by

$$\inf_{T \preceq T_{\max}} R_\alpha(\widehat{Y}(T)), \tag{A.27}$$

where the temperature $\alpha$ is $\Theta((d/n)\log(n/d))$. Note that (A.27) is smaller than the minimum of $R_\alpha(\widehat{Y}(T_K)) = \overline{\mathrm{err}}(\widehat{Y}(T_K)) + \alpha|T_K|$ over all fully grown trees $T_K$ of depth $K$ with $|T_K| \leq 2^K$, i.e.,

$$\inf_{K \geq 1} \{ \overline{\mathrm{err}}(\widehat{Y}(T_K)) + \alpha 2^K \}. \tag{A.28}$$

Combining the training error bound (A.26) with (A.28), we are led to optimize

$$\widehat{\sigma}_Y^2 \Big( 1 - \frac{K}{\log_2(4K^a An)} \Big)^{\widehat{\rho}_{\mathcal{M}}^2} + \alpha 2^K, \tag{A.29}$$

over $K \geq 1$, although suboptimal choices of $K$ will suffice for our purposes. Choosing $K$ to satisfy $K = \lceil \log_2 \big( \frac{\widehat{\sigma}_Y^2 (\log_2(4K^a An))^{-\widehat{\rho}_{\mathcal{M}}^2}}{\alpha} \big) \rceil < \lceil \log_2(\widehat{\sigma}_Y^2/\alpha) \rceil$, we find that (A.29) is equal to

$$\widehat{\sigma}_Y^2 \Big( \frac{\log_2(4K^a An\alpha(\log_2(4K^a An))^{\widehat{\rho}_{\mathcal{M}}^2}/\widehat{\sigma}_Y^2)}{\log_2(4K^a An)} \Big)^{\widehat{\rho}_{\mathcal{M}}^2} + \widehat{\sigma}_Y^2 \Big( \frac{1}{\log_2(4K^a An)} \Big)^{\widehat{\rho}_{\mathcal{M}}^2}$$

$$= \mathcal{O}\Big( \widehat{\sigma}_Y^2 \Big( \frac{\log((d/\widehat{\sigma}_Y^2)\log^{2+a}(n))}{\log(n)} \Big)^{\widehat{\rho}_{\mathcal{M}}^2} \Big).$$

Combining this bound with Theorem 1 proves (23). □

# B  Auxiliary Lemmas

**Lemma B.1.** *Suppose the density of $\mathbf{X}$ never vanishes and $\Delta(s^*, \mathrm{t}) > 0$. Then the conditional probability of the left daughter node along the splitting variable, i.e., $\mathbb{P}(X \le s^* \mid \mathbf{X} \in \mathrm{t})$, has the form*

$$\frac{1}{2} \pm \frac{1}{2}\sqrt{\frac{v}{v + \rho^2(\widehat{Y}^*, Y \mid \mathbf{X} \in \mathrm{t})}}, \tag{B.1}$$

*where $v = \frac{(\mathbb{E}[Y \mid \mathbf{X} \in \mathrm{t},\ X = s^*] - \mathbb{E}[Y \mid \mathbf{X} \in \mathrm{t}])^2}{VAR(Y \mid \mathbf{X} \in \mathrm{t})}$.*

*Proof.* Recall from (4) (albeit, the infinite sample version) that one can write

$$\Delta(s, \mathrm{t}) = P(\mathrm{t}_L) P(\mathrm{t}_R) (\mathbb{E}[Y \mid \mathbf{X} \in \mathrm{t},\ X \le s] - \mathbb{E}[Y \mid \mathbf{X} \in \mathrm{t},\ X > s])^2. \tag{B.2}$$

Next, define

$$\Xi(s) = P(\mathrm{t}_L) P(\mathrm{t}_R) (\mathbb{E}[Y \mid \mathbf{X} \in \mathrm{t},\ X \le s] - \mathbb{E}[Y \mid \mathbf{X} \in \mathrm{t},\ X > s]),$$

so that

$$\Delta(s, \mathrm{t}) = |\Xi(s)|^2 / (P(\mathrm{t}_L) P(\mathrm{t}_R)). \tag{B.3}$$

An easy calculation shows that

$$\frac{\partial}{\partial s} \Xi(s) = p(\mathrm{t}_L)(\mathbb{E}[Y \mid \mathbf{X} \in \mathrm{t},\ X = s] - \mathbb{E}[Y \mid \mathbf{X} \in \mathrm{t}]) = p(\mathrm{t}_L) G(s), \tag{B.4}$$

where $p(\mathrm{t}_L) = \frac{\partial}{\partial s} \mathbb{P}(X \le s \mid \mathbf{X} \in \mathrm{t})$ and $G(s) = \mathbb{E}[Y \mid \mathbf{X} \in \mathrm{t},\ X = s] - \mathbb{E}[Y \mid \mathbf{X} \in \mathrm{t}]$.

Taking the derivative of $\Delta(s, \mathrm{t})$ with respect to $s$, we find that

$$\frac{\partial}{\partial s} \Delta(s, \mathrm{t}) = \frac{\Xi(s) p(\mathrm{t}_L)(2P(\mathrm{t}_L)P(\mathrm{t}_R)G(s) - \Xi(s)(1 - 2P(\mathrm{t}_L)))}{(P(\mathrm{t}_L)P(\mathrm{t}_R))^2}. \tag{B.5}$$

Suppose $s^*$ is a global maximizer of (B.3) (in general, it need not be unique). Then a necessary condition (first-order optimality condition) is that the derivative of $\Delta(s, \mathrm{t})$ is zero at $s^*$. That is, from (B.5), $s^*$ satisfies

$$\Xi(s^*) p(\mathrm{t}_L^*)(2P(\mathrm{t}_L^*)P(\mathrm{t}_R^*)G(s^*) - \Xi(s^*)(1 - 2P(\mathrm{t}_L^*))) = 0, \tag{B.6}$$

where we denote the daughter nodes with an optimal theoretical split $s^*$ by $\mathrm{t}_L^*$ and $\mathrm{t}_R^*$, i.e., $\mathrm{t}_L^* = \{\mathbf{X} \in \mathrm{t} : X \le s^*\}$ and $\mathrm{t}_R^* = \{\mathbf{X} \in \mathrm{t} : X > s^*\}$. By assumption, $p(\mathrm{t}_L^*) > 0$ (since the density of $\mathbf{X}$ never vanishes) and $\Delta(s^*, \mathrm{t}) > 0$. It follows from rearranging (B.6) and using the identity (B.3) that

$$P(\mathrm{t}_L^*) = \frac{1}{2} - \frac{\mathrm{sgn}(\Xi(s^*)) \times G(s^*)}{\sqrt{\Delta(s^*, \mathrm{t})}} \sqrt{P(\mathrm{t}_L^*)P(\mathrm{t}_R^*)}. \tag{B.7}$$

The solution to (B.7) is obtained by solving a simple quadratic equation of the form $p = 1/2 \pm c\sqrt{p(1-p)}$, $0 \le p \le 1$, and noting from Lemma A.1 that $\Delta(s^*, \mathrm{t}) = \Delta(\mathrm{t}) \times \rho^2(\widehat{Y}^*, Y \mid \mathbf{X} \in \mathrm{t})$, which proves the identity (B.1). $\qquad \square$

**Lemma B.2.** *Suppose $X$ is uniformly distributed on the unit interval and $R = \inf\{r \ge 1 : g^{(r)}(\cdot) \text{ exists and is continuous and nonzero at } c\} < \infty$, where $c \in [0, 1]$. Then*

$$\liminf_{(a,b) \to (c,c)} \left\{ \frac{VAR(g(X) \mid X \in [a,b])}{\left( \int_a^b |g'(x)| \sqrt{\frac{x-a}{b-a} \frac{b-x}{b-a}} dx \right)^2} \right\} = \Omega(1/R). \tag{B.8}$$

*Proof.* Since the distribution of $(X - a)/(b - a)$ given $X \in [a, b]$ is uniform on the unit interval, the ratio in the limit infimum (B.8) is

$$\frac{VAR(g(X(b-a) + a))}{\left((b-a)\int_0^1 |g'(x(b-a) + a)| \sqrt{x(1-x)}dx\right)^2}.$$

Let $\delta = (c-a)/(b-a)$. By a Taylor expansion of $g'(\cdot)$ and the definition of $R$, for fixed $\delta$,

$$\lim_{(a,b)\to(c,c)} (b-a)^{-R} \int_0^1 |g'(x(b-a)+a)|\sqrt{x(1-x)}dx \tag{B.9}$$

$$= \frac{|g^{(R)}(c)|}{(R-1)!} \int_0^1 |x-\delta|^{R-1}\sqrt{x(1-x)}dx. \tag{B.10}$$

For the variance, first note that

$$\mathrm{VAR}(g(X(b-a)+a)) = \int_0^1 \left(g(x(b-a)+a) - \int_0^1 g(x'(b-a)+a)dx'\right)^2 dx.$$

Let $D(x)$ denote the divided difference $\frac{g(x(b-a)+a)-g(c)}{(x(b-a)+a-c)^R}$. Then, we can rewrite $(b-a)^{-R}(g(x(b-a)+a) - \int_0^1 g(x'(b-a)+a)dx')$ as

$$D(x)(x-\delta)^R - \int_0^1 D(x')(x'-\delta)^R dx'. \tag{B.11}$$

Next, use a Taylor expansion of $g(\cdot)$ about the point $c$ and continuity of $g^{(R)}(\cdot)$ at $c$ to argue that

$$\lim_{(a,b)\to(c,c)} D(x) = \frac{g^{(R)}(c)}{R!},$$

where the convergence is uniform and the limit is nonzero by definition of $R$. Therefore, for fixed $\delta$,

$$\lim_{(a,b)\to(c,c)} (b-a)^{-2R}\mathrm{VAR}(g(X(b-a)+a)) \tag{B.12}$$

$$= \left(\frac{g^{(R)}(c)}{R!}\right)^2 \int_0^1 \left((x-\delta)^R - \int_0^1 (x'-\delta)^R dx'\right)^2 dx$$

$$= \left(\frac{g^{(R)}(c)}{R!}\right)^2 \mathrm{VAR}((X-\delta)^R). \tag{B.13}$$

Combining (B.9) and (B.13), we have that the limit infimum (B.8) is at least

$$\inf_\delta \frac{\mathrm{VAR}((X-\delta)^R)}{(R\int_0^1 |x-\delta|^{R-1}\sqrt{x(1-x)}dx)^2}. \tag{B.14}$$

Tedious calculations show that the infimum is achieved at $\delta = 1/2$ and hence (B.14) is $\Omega(1/R)$. $\square$

**Lemma B.3.** *Consider the expression (A.22). Then,*

$$\frac{N(\mathrm{t})}{\sum_{k=1}^V (b_{i_k} - b_{i_{k-1}})^2/D_k} \geq \frac{1}{D^{-1}MN(\mathrm{t}) + (V-M-1)\wedge(1+\log(2N(\mathrm{t})))}, \tag{B.15}$$

*where $M$, $V$, and $D$ are defined in Lemma A.4.*

*Proof.* For brevity, we omit dependent on $\mathrm{t}$ and write $N$ instead of $N(\mathrm{t})$.

Suppose that $b_i$ changes sign at index $i_k$ (one of the $M$ many indices such that $b_{i_{k-1}}b_{i_k} < 0$). Then, since $b_{i_k} = \mathrm{sgn}(a_{i_k} - a_{i_{k-1}})\sqrt{i_k(N-i_k)}$, we have

$$\sum_{k:b_{i_{k-1}}b_{i_k}<0} \frac{(b_{i_k} - b_{i_{k-1}})^2}{ND_k} = \sum_{k:b_{i_{k-1}}b_{i_k}<0} \frac{(|b_{i_k}| + |b_{i_{k-1}}|)^2}{ND_k}$$

$$\leq \sum_{k:b_{i_{k-1}}b_{i_k}<0} \frac{(|b_{i_k}| + |b_{i_{k-1}}|)^2}{ND}$$

$$\leq D^{-1}MN,$$

where the last line is from $(|b_{i_k}| + |b_{i_{k-1}}|)^2 = (\sqrt{i_k(N-i_k)} + \sqrt{i_{k-1}(N-i_{k-1})})^2 \leq N^2$. Next, for the remaining $V - M$ indices such that $b_{i_{k-1}}b_{i_k} > 0$ we have,

$$\sum_{k:b_{i_{k-1}}b_{i_k}>0} \frac{(|b_{i_k}| - |b_{i_{k-1}}|)^2}{ND_k} \leq \sum_{k:b_{i_{k-1}}b_{i_k}>0} \frac{|N - i_k - i_{k-1}|}{N}$$
$$\leq V - M - 1,$$

where the last line follows from the fact there is always one index such that $|N - i_k - i_{k-1}| + |N - i_{k+1} - i_k| = |i_{k+1} - i_{k-1}|$, namely, at $k^* := \min\{k : i_k + i_{k-1} \geq N\}$. Thus, it follows that $\frac{N}{\sum_{k=1}^{V}(b_{i_k} - b_{i_{k-1}})^2/D_k}$ is at least

$$\frac{1}{D^{-1}MN + (V - M - 1) \wedge \sum_{k=1}^{V} \frac{(|b_{i_k}| - |b_{i_{k-1}}|)^2}{ND_k}}. \tag{B.16}$$

We now obtain an upper bound for

$$\sum_{k=1}^{V} \frac{(|b_{i_k}| - |b_{i_{k-1}}|)^2}{ND_k} = \sum_{k=1}^{V} \frac{D_k(N - i_k - i_{k-1})^2}{N(\sqrt{i_k(N-i_k)} + \sqrt{i_{k-1}(N-i_{k-1})})^2}. \tag{B.17}$$

Now, $(\sqrt{i_k(N-i_k)} + \sqrt{i_{k-1}(N-i_{k-1})})^2 \geq (2N - i_k - i_{k-1})(i_k + i_{k-1} - N)$ for all $k \geq k^*$. Thus, the sum $\sum_{k \geq k^*} \frac{D_k(N-i_k-i_{k-1})^2}{N(\sqrt{i_k(N-i_k)} + \sqrt{i_{k-1}(N-i_{k-1})})^2}$ is at most

$$\sum_{k \geq k^*} \frac{D_k}{2N - i_k - i_{k-1}}\left(\frac{i_{k-1} + i_k}{N} - 1\right) \leq \sum_{k \geq k^*} \frac{i_k - i_{k-1}}{2N - i_k - i_{k-1}}, \tag{B.18}$$

where we used the fact that $D_k = i_k - i_{k-1}$. Next, $(\sqrt{i_k(N-i_k)} + \sqrt{i_{k-1}(N-i_{k-1})})^2 \geq (i_k + i_{k-1})(N - i_k - i_{k-1})$ for all $k < k^*$ and hence the sum $\sum_{k<k^*} \frac{D_k(N-i_k-i_{k-1})^2}{N(\sqrt{i_k(N-i_k)} + \sqrt{i_{k-1}(N-i_{k-1})})^2}$ is at most

$$\sum_{k<k^*} \frac{D_k}{i_k + i_{k-1}}\left(1 - \frac{i_{k-1} + i_k}{N}\right) \leq \sum_{k<k^*} \frac{i_k - i_{k-1}}{i_k + i_{k-1}}. \tag{B.19}$$

Combining (B.18) and (B.19), we have shown that (B.17) is at most

$$\sum_{k<k^*} \frac{i_k - i_{k-1}}{i_k + i_{k-1}} + \sum_{k \geq k^*} \frac{i_k - i_{k-1}}{2N - i_k - i_{k-1}}. \tag{B.20}$$

The sum (B.20) is largest when $V = N$, yielding

$$\sum_{i=1}^{(N-1)/2} \frac{1}{2i - 1} + \sum_{i=1}^{(N+1)/2} \frac{1}{2i - 1} \leq 1 + \log(2N). \tag{B.21}$$

Combining (B.20) and (B.21) with (B.16) proves (B.15). $\qquad\square$

## Footnotes

[2]More precisely, if $I_1, \ldots, I_M$ are the stationary intervals of $g(\cdot)$ and $D_k = \#\{X_i \in I_k\}$, then $D = \min_k\{D_k : D_k \geq 1\}$.