[Reviews · NeurIPS 2020]

Review 1

Summary and Contributions: This paper studies the CART algorithm with a new point of view based on Pearson correlation. They connect several parts of the algorithm to this notion of correlation which allow to study (asymptotic and finite case) consistency for sparse models. Following the author's response: I thank the authors for the newly added section. It is definitely a good and interesting point to extend to ensemble of trees / random forests.

Strengths: The main interest of this work is the new way of studying the CART algorithm. To the best of my knowledge, this was not done using Pearson correlation and this is of interest to encourage researchers to use alternative ways to show consistency or other desirable properties of tree-based models.

Weaknesses: It is pointed in the conclusion but I think it is still the main weakness of this work. This work focuses only on CART for which its soundness is already quite well established and it is very hard to see (or to be convinced) that such results are useful for methods made of ensemble of randomised trees that are used in practice nowadays (and sometimes without optimal split search). I would like a more in-depth discussion than the classical principle of ensemble (that is better in average than a single decision tree).

Correctness: Yes, I think so.

Clarity: Yes.

Relation to Prior Work: Yes.

Reproducibility: Yes

Additional Feedback: In appendix, some refs are not well compiled (or missing).


Review 2

Summary and Contributions: UPDATE: I read the rebuttal and I believe the added experiments clearly improve the paper. The paper analyzes the statistical properties of splits for regression decision trees (DT) and find that decision splits are related to Pearson correlation. Specifically the pearson correlation of the split output and response variable, influences the location of the split (the higher the correlation, the better the split) and the error reduction in the tree. This is specially relevant in sparse problems in which the number of informative variables is much smaller than the number of variables (d0<<d). For this cases, the decision trees are consistent, which is not possible with other methods such as K-NN or kernel methods without a prior dimensionality reduction.

Strengths: The paper seems sound in their derivations and, as far as I am aware, novel.

Weaknesses: From my point of view I would include an empirical evaluation of some sort to illustrate the main results. Even using ad-hoc synthetic datasets would be of interest. Another aspect that is unclear to me is what are the consequences of "Suppose X is uniformly distributed" in the results since the data is not generally uniformly distributed.

Correctness: There is no empirical methodology as the paper is theoretical. Regarding the theorems they seem to be correct but I did not fully revise them

Clarity: The paper is clear although I would carry out a proof reading of it as there are some typos.

Relation to Prior Work: Yes. Another reference that may be of interest is: A. B. Nobel, "Analysis of a complexity-based pruning scheme for classification trees," in IEEE Transactions on Information Theory, vol. 48, no. 8, pp. 2362-2368, Aug. 2002, doi: 10.1109/TIT.2002.800482.

Reproducibility: Yes

Additional Feedback:


Review 3

Summary and Contributions: This paper sought to investigate theoretical and statistical properties of the CART methodology that are often overlooked. It provides an in-depth explanation and analysis of a CART algorithm, allowing others to replicate it. In doing so, it proves the reduction in training error among every recursive binary split. Additionally, models trained with this methodology have a high probability of having a bounded training error, even with arbitrary response sparsity. Proving the effectiveness of the model is important to anyone who is seeking to make a data-dependent decision. It also sheds light on the dependency decision stumps have on the most impactful data. Where other algorithms may have a stronger bias because of the assumption that all data is relevant, models following the CART methodology are less susceptible to approximation error because the recursive splits are based on the most relevant data. This helps any data-driven decision makers compare and contrast different supervised learning algorithms.

Strengths: see above

Weaknesses: see above

Correctness: see above

Clarity: Overall, I think this paper is well written because of its thorough and intuitive structure. However, it seems difficult for readers without expertise in the subject area to follow along with the mathematical notation. The summaries following the sections with mathematical notation serve as a good supplement for a complete understanding of the paper.s

Relation to Prior Work: see above

Reproducibility: Yes

Additional Feedback:


Review 4

Summary and Contributions: This paper demonstrates that the training error of CART is governed by the Pearson correlation between the optimal decision stump and response data in each node. Based on this observation, the authors achieve an optimal complexity/goodness-of-fit trade-off and prove that the convergence rate of prediction errors is controlled by data dependent quantities.

Strengths: -There are solid analyses on the derivation of the connection between the Pearson correlation and training error, as well as the convergence of prediction errors. -The interpretability of the decision tree algorithms is demonstrated with thorough mathematical proof.

Weaknesses: -Given that the scope of this paper, as indicated in the title, is high-dimensional and sparse learning with CART, I would like to see a concrete performance evaluation of the proposed regression model on real datasets in addition to the theoretical deduction. -In Assumption 1, it is not clear why the quantity governs the convergence rate of both the training error and prediction. -In Ln. 56, the authors claim that the training error is bounded instead of designing the proxy task for approximation error, but the motivation is not clear and detailed to me. The authors' rebuttal provides detailed feedback on my concerns about the training error and convergence. The added experiments of comparing with simple KNN improve the paper to some extent.

Correctness: The methodology is correct in this paper.

Clarity: Most of the paper is clear and well written.

Relation to Prior Work: Yes.

Reproducibility: Yes

Additional Feedback: See the weakness.

[Author Response · NeurIPS 2020]

We thank each reviewer for their insightful and constructive feedback. It will surely improve the manuscript.

**Experiments.** Reviewers #2 and #4 suggested that we illustrate our theory with experiments. We wholeheartedly agree. Following this suggestion, our paper now includes experiments with various synthetic/real-world datasets and compares CART with $k$-NN and other kernel methods. In Fig. 1, we show the outcome of one such experiment. We sample $\{(\mathbf{X}_i, Y_i)\}_{1 \le i \le n}$ i.i.d. with $n = 1000$ and consider a sparse additive model $Y = \sum_{j=1}^{d_0} g_j(X_j)$ with $d_0 = 5$ component functions, where each $g_j(X_j)$ equals $\pm X_j^2$ (alternating signs) and $\mathbf{X} \sim \text{Uniform}([0,1]^d)$. We then plot the test error of CART vs. $k$-NN as $d$ ranges from 5 to 100. According to Theorems 3 and 4, the convergence rate of CART depends primarily on the sparsity $d_0$ and therefore its performance should not be adversely affected by growing $d$. Consistent with our theory, the prediction error of CART remains stable as $d$ increases, whereas $k$-NN does not adapt to the sparsity.

**Figure 1:** Prediction error (averaged over 10 independent replications) of pruned CART vs. $k$-NN (with cross-validated $k$) as a function of ambient dimension $d \in \{d_0, \dots, 100\}$ with fixed sparsity $d_0 = 5$. CART is impervious to increasing ambient dimensionality, whereas $k$-NN suffers and does not adapt to sparsity.

**Distributional assumptions.** Reviewer #2 inquired about the consequences of assuming the input variable $\mathbf{X} = (X_1, \dots, X_d)$ is uniformly distributed on $[0,1]^d$. For independent predictor variables $X_j$, there is no loss of generality in assuming uniform marginal distributions. Indeed, CART trees are invariant to strictly monotone transformations of the individual predictor variables. One such transformation is the marginal cumulative distribution function $F_{X_j}(\cdot)$ of the predictor variables, for which $F_{X_j}(X_j) \sim \text{Uniform}([0,1])$—and so the problem can be reduced to the uniform case. For dependent predictor variables, the proofs go through if the joint density of $\mathbf{X}$ is bounded above and below by a positive constant, though the convergence rates have worse dependence on the sparsity level.

**Connection to ensemble models.** Reviewer #1 would like to see a more in-depth discussion of how the analysis for CART trees can be carried over to random forests—one that goes beyond the ensemble principle. This is an excellent point and we agree that it deserves more attention. Therefore, we have included a new section that explicitly describes how our results can be used to show analogous adaptivity properties for random forests. Let us briefly mention that, while the efficacy of randomization in random forests (e.g., bagging or random feature selection) is not fully understood from a theoretical perspective, basic properties such as consistency often begin with a study of the individual trees [Scornet et al., 2015, Wager & Athey, 2017]. Finally, it is indeed true that the empirical soundness of CART has been well-documented over the past 30+ years—however—many of its theoretical properties (like adaptivity to sparsity) have not been made rigorous. Since CART is still widely used for its simplicity and interpretability, the primary goal of this paper is to put these empirical observations on a solid theoretical foundation.

**Proxy for estimation and training error.** Reviewer #4 asked for further clarification on how we avoid using the node diameters as a proxy for the approximation error and, instead, directly bound the training error. The extant approach typically bounds the expected prediction error (over the training data) by analyzing the approximation and estimation errors separately (the estimation error is usually less troublesome). Because CART outputs the average of the response values in a node, if the regression function is smooth, the approximation error is at most a constant multiple of the largest node diameter. Thus, one can use the node diameters as a proxy for the approximation error. In contrast, we use the fact that the prediction error is with high-probability (over the training data) bounded by the training error plus a complexity term. The connection between the Pearson correlation and the training error (see Lemma 1) facilitates our analysis and allows us to prove more fine-grained results.

**Quantity that governs the training and prediction errors.** Reviewer #4 asked for clarification on whether the quantity in Assumption 1 governs the convergence rate of both the training error and prediction error. Assumption 1 is merely a technical condition about the maximum number of data points in each node and enables one to use the correlation comparison inequality in Fact 2—it does not explicitly control the training and prediction errors. As mentioned in the discussion preceding Theorem 4, it is $\widehat{\rho}_{\mathcal{M}}$ (i.e., the largest correlation between the response data and a monotone function of an individual predictor variable) that explicitly controls the rate at which both errors tend to zero.

**Additional reference.** Reviewer #2 mentioned the reference [Nobel, 2002]. We thank the reviewer for pointing this out to us and have cited it in a revised version of the paper.

**Accessibility to broader audience.** Reviewer #3 expressed concern that the paper may seem difficult for readers without expertise in the subject area to follow along with the mathematical notation. We have carefully gone through the paper to explain and write all notation in a more accessible way that respects the reader's background. We have also carried out a thorough proof-reading to correct any typos or references that were not properly compiled (as mentioned by Reviewers #1 and #2).

[Meta-Review · NeurIPS 2020]

The reviewers generally agreed that this was a well-written and comprehensive paper that makes a valuable contribution. While there was some uncertainty about how significantly this work will motivate future research, this was not considered to be a strongly negative point. The main concerns were the lack of experiments (R2 and R4), and the limited discussion of ensembles (R1). The author response did a great job in addressing these concerns: all three of these reviewers seem to be mostly satisfied with the promised changes. Overall, this is a solid paper, and the changes proposed in the author response should make it even better. In terms of smaller issues, please carefully read the reviews, and take their suggestions seriously when making edits.